# THE GENERALIZATION GAP IN OFFLINE REINFORCEMENT LEARNING

**Ishita Mediratta**[*α] **Qingfei You**[*α] **Minqi Jiang** [α β] **Roberta Raileanu** [α β]

[α] Meta, [β] University College London

## ABSTRACT

Despite recent progress in offline learning, these methods are still trained and tested on the same environment. In this paper, we compare the generalization abilities of widely used online and offline learning methods such as online reinforcement learning (RL), offline RL, sequence modeling, and behavioral cloning. Our experiments show that offline learning algorithms perform worse on new environments than online learning ones. We also introduce the first benchmark for evaluating generalization in offline learning, collecting datasets of varying sizes and skill-levels from Procgen (2D video games) and WebShop (e-commerce websites). The datasets contain trajectories for a limited number of game levels or natural language instructions and at test time, the agent has to generalize to new levels or instructions. Our experiments reveal that existing offline learning algorithms struggle to match the performance of online RL on both train and test environments. Behavioral cloning is a strong baseline, outperforming state-of-the-art offline RL and sequence modeling approaches when trained on data from multiple environments and tested on new ones. Finally, we find that increasing the diversity of the data, rather than its size, improves performance on new environments for all offline learning algorithms. Our study demonstrates the limited generalization of current offline learning algorithms highlighting the need for more research in this area.

## 1 INTRODUCTION

Training agents offline from static datasets (Levine et al., 2020; Reed et al., 2022; Lee et al., 2022) has demonstrated significant potential in application domains where online data collection can be expensive or dangerous, such as healthcare (Liu et al., 2020a), education (Singla et al., 2021), robotics (Singh et al., 2022), or autonomous driving (Kiran et al., 2021; Prudencio et al., 2023). The abillity to generalize to new scenarios is crucial for the safe deployment of these methods particularly in such high-stakes domains. However, the generalization abilities of offline learning algorithms to new environments (with different initial states, transition functions, or reward functions) remains underexplored. A key reason for this is that existing offline learning datasets predominantly focus on singleton environments where all trajectories are from the same environment (such as playing an Atari game or making a humanoid walk), thereby limiting the evaluation of generalization. In contrast, there is a large body of work focused on evaluating and improving the generalization of online reinforcement learning (RL) methods Cobbe et al. (2018); Packer et al. (2019); Zhang et al. (2018a;b); Cobbe et al. (2020); Küttler et al. (2020); Raileanu et al. (2021); Raileanu & Fergus (2021); Jiang et al. (2021a). This paper aims to bridge this gap by assessing the generalization performance of offline learning algorithms (including behavioral cloning, sequence modeling (Chen et al., 2021), and state-of-the-art offline RL (Fujimoto et al., 2018; 2019; Kumar et al., 2020a; Kostrikov et al., 2020) approaches) in two different scenarios: (1) *unseen levels* in the case of Procgen (Cobbe et al., 2020) and (2) *unseen instructions* in the case of WebShop (Yao et al., 2022). Our results show that none of the benchmarked offline learning methods, (i.e. BCQ (Fujimoto et al., 2018), CQL (Kumar et al., 2020a), IQL (Kostrikov et al., 2021), BCT, and DT (Chen et al., 2021)) are able to generalize as well as behavioral cloning (BC), underscoring the need of developing offline learning methods with better generalization capabilities.

---

[*]These authors contributed equally to this work

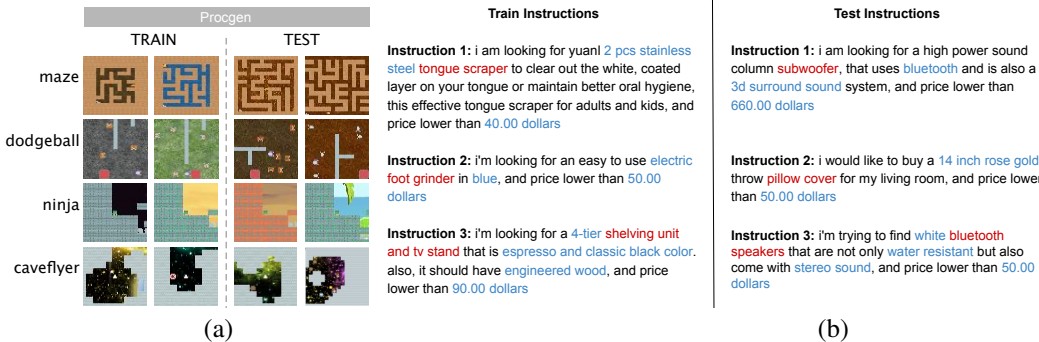

Figure 1: (a) Sample screenshots from the train and test environments of four Procgen games. (b) Sample instructions (item descriptions) from the train and test set of human demonstrations from WebShop. Red and blue highlight the type and attributes of the desired item, respectively.

In this work, we first introduce a collection of offline learning datasets of different sizes and skill-levels from Procgen (Cobbe et al., 2020) and WebShop (Yao et al., 2022) to facilitate a comprehensive evaluation of the generalization abilities of offline learning algorithms. The Procgen benchmark consists of 16 procedurally generated 2D video games which differ in their visual appearances, layouts, dynamics, and reward functions. Since the levels are procedurally generated, generalization to new levels can be assessed in this benchmark. We create a number of Procgen datasets that aim to test an agent's ability of *solving new levels* i.e., with the same reward function but different initial states and dynamics.

Webshop is a simulated e-commerce website environment with more than 1 million real-world products. Given a text instruction describing the desired product, the agent needs to navigate multiple types of webpages and issue different actions to find, customize, and purchase an item. We create a number of WebShop datasets both by using the human demonstrations provided and by generating suboptimal trajectories, which aim to test an agent's ability of *following new instructions* i.e., with the same dynamics but different initial states and reward functions. Figure 1 shows some sample observations from Procgen and trajectories from WebShop.

We then benchmark a variety of widely-used offline learning algorithms, allowing us to verify the generalization of these algorithms and establish baselines for future research. On the expert and suboptimal datasets from Procgen (Section 4.1), all offline learning methods underperform online RL at test time, with BC generally outperforming offline RL and sequence modeling approaches. On WebShop's human demonstrations dataset (Section 4.6), BC again outperforms the offline learning methods in terms of both final score and success rate. We also study the generalization of these algorithms as the diversity (Section 4.4) and size (Section 4.5) of the training data increases, observing that an increase in data diversity significantly improves generalization while increasing the size of training data does not. These findings not only provide insights into the strengths and weaknesses of existing algorithms but also emphasize the necessity for more research on understanding and improving generalization in offline learning. We hope our open-sourced datasets, baseline implementations, and proposed evaluation protocols can help lower the barrier for future research in this area.

## 2 BACKGROUND

This work studies the effectiveness of offline learning algorithms in *contextual Markov decision processes* (CMDPs; Hallak et al., 2015). A CMDP is defined as a tuple $\mathcal{M} = (\mathcal{C}, \mathcal{S}, \mathcal{A}, \mathcal{M}(c))$, where $\mathcal{C}$ is the set of contexts, $\mathcal{S}$ is the set of states, $\mathcal{A}$ is the set of actions and $\mathcal{M}$ is a function that maps a specific context $c \in \mathcal{C}$ to a Markov decision process (Sutton & Barto, 2018, MDP) $\mathcal{M}(c) = (\mathcal{S}, \mathcal{A}, \mathcal{T}^c, \mathcal{R}^c, \rho^c)$, where $\mathcal{T}^c : \mathcal{S} \times \mathcal{A} \times \mathcal{C} \rightarrow S$ is the contextual transition function, $\mathcal{R}^c : \mathcal{S} \times \mathcal{C} \rightarrow \mathbb{R}$ is the contextual reward function, and $\rho^c$ is the initial state distribution conditioned on the context. Given a CMDP, reinforcement learning (RL) seeks to maximize the value of the agent's policy $\pi$ defined as $\mathbb{E}\left[\sum_{t=0}^{T} r_t \gamma^t\right]$, where $r_t$ is the reward at time $t$ and $T$ is the time horizon.

Note that $c \in \mathcal{C}$ is not observable. The space of contexts $\mathcal{C}$ will be split into a training set $\mathcal{C}_{train}$, a validation set $\mathcal{C}_{val}$, and a test set $\mathcal{C}_{test}$. The training contexts are used for generating the datasets, the validation ones are used for performing hyperparameter sweeps and model selection, and the

test sets are used to evaluate the agents via online interactions with the environment. In the case of Procgen, the context $c$ corresponding to an instance of the environment (or level) an $\mathcal{M}(c)$ determines the initial state and transition function. In the case of Webshop, the context $c$ corresponding to an instance of the environment (or instruction) $\mathcal{M}(c)$ determines the initial state and reward function.

In this paper, the agents *learn from an offline dataset $\mathcal{D}$* which contains trajectories generated by a behavior policy $\pi_B$ by interacting with the training contexts. This policy can have different degrees of expertise (e.g., expert or suboptimal), hence the offline learning algorithms must learn to extract meaningful behaviors using this static dataset without access to any online interactions with the environment. Since the dataset is fixed, it typically does not cover the entire state-action distribution of the environment. Because of this, offline RL algorithms can suffer from distributional shift and hence, they must employ techniques that enable them to generalize to new states at test time (Levine et al., 2020) or prevent sampling actions which are out-of-distribution (Kumar et al., 2020a; Fujimoto et al., 2018; Kostrikov et al., 2021).

## 3 EXPERIMENTAL SETUP

For Procgen, we use $|\mathcal{C}_{train}| = 200$, $|\mathcal{C}_{val}| = 50$, and $|\mathcal{C}_{test}| = 100$, in line with prior work (Cobbe et al., 2020; Raileanu & Fergus, 2021; Jiang et al., 2021b), while for WebShop we use $|\mathcal{C}_{train}| = 398$, $|\mathcal{C}_{val}| = 54$, and $|\mathcal{C}_{test}| = 500$ instructions, unless otherwise noted (Yao et al., 2022). We collect the following datasets as part of our offline learning benchmark: **(1) Procgen expert dataset with 1M transitions, (2) Procgen mixed expert-suboptimal dataset with 1M transitions, (3) Procgen expert dataset with 10M transitions, (4) Procgen suboptimal with 25M transitions, (5) WebShop human dataset with 452 trajectories, and (6) WebShop suboptimal datasets with 100, 1K, 1.5K, 5K, and 10K trajectories.**

For the Procgen datasets, we evaluate 7 methods which are competitive on other offline learning benchmarks (Kurin et al., 2017; Fu et al., 2020; Agarwal et al., 2020; Todorov et al., 2012) and frequently used in the literature (Fu et al., 2020; Gulcehre et al., 2020; Qin et al., 2022): (1) **Behavioral Cloning (BC)**, (2) **Batch Constrained Q-Learning (BCQ) Fujimoto et al. (2019)**, (3) **Conservative Q-Learning (CQL) Kumar et al. (2020a)**, (4) **Implicit Q-Learning (IQL) Kostrikov et al. (2021)**, (5) **Behavioral Cloning Transformer (BCT) Chen et al. (2021)**, and (6) **Decision Transformer (DT) Chen et al. (2021)**. For the WebShop datasets, we evaluate BC, CQL and BCQ. We cannot evaluate existing transformer-based approaches such as DT or BCT due to their limited context lengths of the underlying causal transformer.. Many WebShop states have 512 tokens, so we typically cannot fit multiple (state, action) pairs in the transformer's context. Similarly for IQL, it is not straightforward to implement the loss function since in WebShop, the action space differs for each state so the size of the action space is not fixed. Since these algorithms cannot be applied to WebShop without significant changes, they are out-of-scope for this paper.

## 4 EXPERIMENTAL RESULTS

### 4.1 GENERALIZATION TO NEW ENVIRONMENTS USING EXPERT DATA

Figure 2 shows the IQM performance (Agarwal et al., 2021b) of baselines averaged across all 16 Procgen games when trained using the 1M expert dataset, normalized using the min-max scores provided in Cobbe et al. (2020). As we can see, *BC outperforms all other sequence modeling or offline RL methods by a significant margin on both train and test levels.* This is in line with prior work which also shows that BC can outperform offline RL algorithms when trained on expert trajectories (Levine et al., 2020). Sequence modeling approaches like DT and BCT perform better than offline RL methods on the training environments, but similarly or slightly worse on the test environments. The gap between BC and DT or BCT is small for training, but large for test. This indicates that, *relative to standard BC, transformer-based policies like DT or BCT, may struggle more with generalization to new scenarios, even if they are just as good on the training environments.* For per-game performance, refer to Figures 21, 22 and Table 5 in Appendix N.

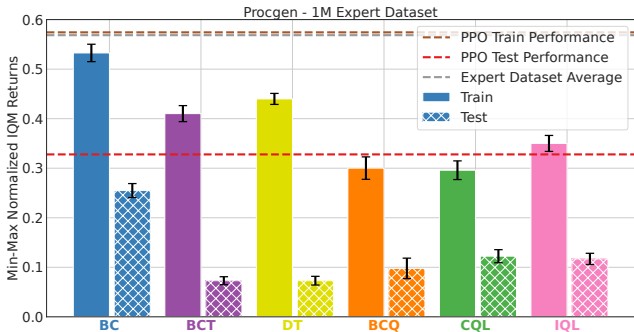

Figure 2: **Performance on Procgen 1M Expert Dataset.** Train and test min-max normalized returns aggregated across all 16 Procgen games, when trained on expert demonstrations. Each method was evaluated online across 100 episodes on levels sampled uniformly from the test set. The IQM aggregate metric is computed over 5 model seeds, with the error bars representing upper (75th) and lower (25th) interval estimates. BC outperforms all offline RL and sequence modelling approaches on both train and test environments. All offline learning methods lag behind online RL on both train and test.

> **Generalization to New Environments (Expert Dataset)**
>
> Existing offline learning methods struggle to generalize to new environments, underperforming online RL at test time even when trained on expert demonstrations (generated by fully-trained online RL agents). Behavioral cloning is a competitive approach, outperforming state-of-the-art offline RL and sequence modeling methods on both train and test environments.

## 4.2 GENERALIZATION TO NEW ENVIRONMENTS USING MIXED EXPERT-SUBOPTIMAL DATA

In the previous section, we observed that offline RL methods struggle to generalize to new environments when trained on expert demonstrations. Prior works (Bhargava et al., 2023; Kumar et al., 2020a; Kostrikov et al., 2021) on singleton environments (where agents are trained and tested on the same environment) show that state-of-the-art offline RL methods typically outperform BC when trained on suboptimal demonstrations. In this section, we investigate whether this finding holds when agents are trained and tested on different environments. For this, we create a *mixed, expert-suboptimal dataset* by uniformly mixing data from the expert PPO checkpoint and another checkpoint whose performance was 50% that of the expert. Therefore, these datasets have average episodic returns of about 3/4th those of the expert datasets.

Contrary to prior results on singleton environments, Figure 3 shows that even with suboptimal data, BC outperforms other offline learning baselines on test levels. However, all methods have a similar generalization gap on average (Figure 3 from Appendix N), suggesting that their generalization abilities are similar. This result indicates that BC can train better on diverse datasets containing trajectories from different environments relative to other offline learning approaches, even if these demonstrations are subotpimal. In Procgen and other CMDPs, it is common for methods with better training performance to also have better test performance since learning to solve all the training tasks is non-trivial and existing algorithms are typically underfitting rather than overfitting. Hence, in such settings the challenges lie both in optimization and generalization (Jiang et al.).

IQL, which achieved state-of-the-art on other singleton environments (Kostrikov et al., 2021), struggles to train well on data from multiple levels and also fails to generalize to new levels at test time. Thus, it appears that training on more diverse datasets with demonstrations from different environments and generalizing to new ones poses a significant challenge to offline RL and sequence modeling approaches, despite their effectiveness when trained on more homogenous datasets from a single environment. However, this finding is not necessarily surprising since these offline learning methods have been developed on singleton environments and haven't been evaluated on unseen environments (different from the training ones). Hence, we believe the community should focus more on the setting we propose here (testing agents in different environments than the ones used to collect training data) to improve the robustness of these algorithms and make them better suited for real-world applications where agents are likely to encounter new scenarios at test time.

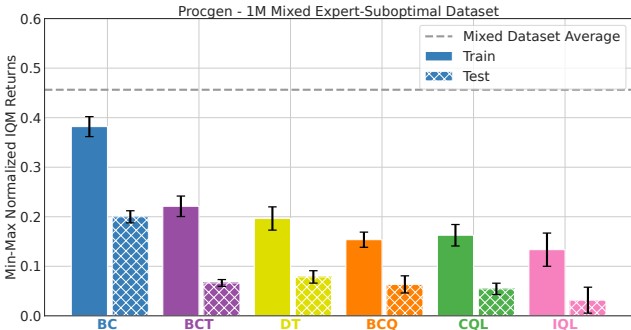

Figure 3: **Performance on Procgen 1M Mixed Expert-Suboptimal Dataset.** Train and test min-max normalized returns aggregated across all 16 Procgen games, when trained on mixed expert-suboptimal demonstrations. Each method was evaluated online across 100 episodes on levels sampled uniformly from the test set. The IQM aggregate metric is computed over 3 model seeds, with the error bars representing upper (75th) and lower (25th) interval estimates. Similar to Figure 2, BC outperforms all offline RL and sequence modelling approaches on both train and test environments. All offline learning methods lag behind online RL on both train and test.

> **Generalization to New Environments (Mixed Expert-Suboptimal Dataset)**
>
> Behavioral cloning also outperforms state-of-the-art offline RL and sequence modeling methods on both train and test environments when learning from suboptimal data from multiple environments.

## 4.3 TRAINING AND TESTING ON A SINGLE ENVIRONMENT

In the previous section, we found that, when trained on data form multiple environments and tested on new ones, BC outperforms offline RL algorithms. However, prior work showed that offline RL methods typically outperform BC when trained on suboptimal data from a single environment and tested on the same one (Kumar et al., 2020a; 2022; Bhargava et al., 2023). At the same time, BC has been shown to outperform offline RL when trained on expert data from a single environment. Here, we aim to verify whether these two observations hold in Procgen in order to confirm the correctness of our implementations.

Therefore, in this section, we show the results when training and testing agents on expert and suboptimal data from a single level. We conduct this experiment across two different datasets with either expert or suboptimal demonstrations, two different game levels with seeds 40 and 1, and all 16 Procgen games. We collect 100,000 expert trajectories in both of these levels by rolling out the final PPO checkpoint, and 100,000 suboptimal trajectories in both of these levels by uniformly sampling out the transitions from two checkpoints, the final one and another checkpoint whose performance is 50% that of the final checkpoint, similar to what we did in the previous section.

Figure 4 shows the performance of these baselines when trained on the expert and suboptimal datasets from level 40. In Appendix I, we report aggregate performance on the expert and suboptimal datasets, as well as per-game performance on both datasets on all 16 games. Figure 4 (top) shows the results when training on the expert dataset, where on most games offline learning methods perform about as well as PPO, with many of these algorithms achieving the maximum score possible (see Coinrun, Leaper, Ninja, Maze, Miner in Figure 10 from Appendix I). Figure 4 (bottom) shows the results when training on the suboptimal dataset, where offline RL methods are either comparable to or better than BC, which is in line with prior work. One exception is Chaser where most offline learning methods struggle to match the average dataset performance when learning from both expert and suboptimal demonstrations. This is not surprising since Chaser is a highly stochastic environment containing multiple moving entities whose actions cannot be predicted. In such settings, one typically needs much more data to learn good policies offline (in order to cover a larger fraction of the states that can potentially be visited). The same applies for other games where the performance of offline learning lags behind that of online learning methods when trained and tested on the same environment. Prior work also shows that offline RL and sequence modeling approaches can struggle to learn robust policies in stochastic environments (Bhargava et al., 2023; Brandfonbrener et al., 2022; Ostrovski et al., 2021; Ghugare et al., 2024).

We report the results on level 1 in Appendix I. These results are consistent with the broader literature which shows that offline RL performs comparable to, or in some case, better than BC when trained and tested on suboptimal demonstrations from the same environment. However, as shown in previous

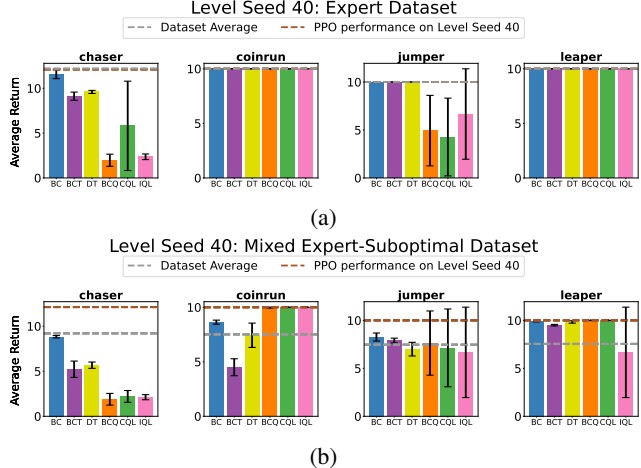

(a)

(b)

Figure 4: Performance of each baseline across selected Procgen games when **trained and tested on the same level using expert and suboptimal dataset**. Here we report performance on selected levels: Chaser, Coinrun, Jumper, and Leaper. For all games, refer to Figures 9 and 10 in Appendix I.

sections, this finding does not hold true when these algorithms are trained and tested on multiple different environments. In such settings, BC tends to outperform other offline learning methods as shown here.

> **Training and Testing on a Single Environment**
>
> All offline learning algorithms perform well when trained and tested in the same environment but struggle to learn and generalize when trained on multiple environments and tested on new ones. When trained and tested on the same environment using expert data, behavioral cloning performs best, as expected. When trained on mixed expert-suboptimal data, offline RL performs comparable to or better than behavioral cloning on most games, which is in line with prior work.

## 4.4 THE EFFECT OF DATA DIVERSITY ON GENERALIZATION

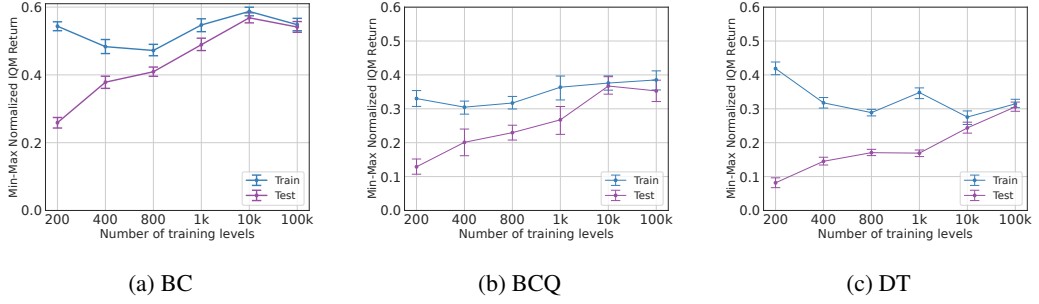

(a) BC          (b) BCQ          (c) DT

Figure 5: **The Effect of Data Diversity on Performance.** Train and test performance of offline learning algorithms for varying number of training levels in the 1M expert datasets, aggregated across all Procgen games. The plot shows the IQM and error bars represent the 75-th and 25th percentiles computed over 3 model seeds. While the training performance doesn't change much with the number of training levels, the test performance increases (and generalization gap decreases) with the diversity of the dataset.

To investigate the role of data diversity on the generalization capabilities of offline learning algorithms, we conduct an experiment to analyze how the performance of each offline learning algorithm scales with the number of training levels while keeping the dataset size fixed to 1M transitions. We run these experiments on Procgen. We consider 200, 400, 800, 1k, 10k and 100k training levels. For each game, we train PPO policies for 25M steps on the corresponding number of levels. We then use the final PPO checkpoints (after 25M training steps) to collect a total of 1M transitions (from the corresponding levels) and train each offline learning algorithm on these datasets (using the same hyperparameters for all datasets). To evaluate these policies, we follow the procedure outlined in Section H.2.1. More specifically, for each dataset we randomly sample 100 test levels from $[n, \infty)$ where $n \in [250, 450, 850, 1050, 10050, 100050]$, respectively, and evaluate the models via online

interactions with these levels. In each case, the levels from $[n - 51, n - 1]$ are used for evaluation and the remaining ones from $[0, n - 51]$ are used for training. Since we use a fixed number of transitions for all datasets, the number of levels is a proxy for the dataset diversity. Figure 5 shows that:

> **The Effect of Data Diversity on Generalization**
>
> Increasing the diversity of the dataset by, for example, increasing the number of training environments while keeping the size of the dataset fixed, leads to significant performance improvements on new environments for all offline learning algorithms.

This finding suggests that in order to train better-generalizing agents using offline data, we should create diverse datasets covering a broad range of experiences. Note that the results presented here use 50 levels for validation for each run. We also experiment with a proportional number of validation levels (with respect to the number of training levels) in Appendix K and find that it leads to the same conclusion.

### 4.5 THE EFFECT OF DATA SIZE ON GENERALIZATION

In other domains of machine learning, it is well-established that dataset size can play a pivotal role in model performance and generalization (Kumar et al., 2022; Kaplan et al., 2020). While the significance of the data diversity has been underscored in the previous section, the impact of the dataset size remains an open question. In this section, we investigate how generalization correlates with the dataset size when the diversity and quality of the dataset is fixed. For this, we scale the training datasets, both expert and suboptimal (in Appendix J), in Procgen by progressively increasing the dataset size from 1 million to 5 million and subsequently to 10 million transitions. Throughout this scaling process, we keep all other hyperparameters same as well as the number of training levels constant at 200. As can be seen in Figure 6, across four offline learning algorithms, there is only a slight increase in both train and test performance as

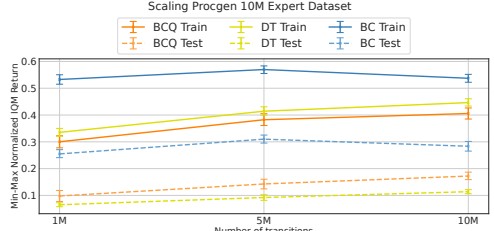

Figure 6: **The Effect of Data Size on Performance (Expert)**. Train and test min-max normalized IQM scores for BC, BCQ and DT as the size of the dataset is increased from 1 to 10 million transitions aggregated across 5 random trials. Scaling the dataset size while keeping the number of training levels fixed (to 200) leads to a slight increase in both train and test returns, but the generalization gap remains about the same. For remaining algorithms, see Figure 16 in Appendix.

a consequence of increasing the dataset size. However, note that the generalization gap remains almost constant. This indicates that increasing the diversity of the dataset (while maintaining its total size) can lead to a bigger generalization improvement than increasing the size of the dataset (while maintaining its diversity) (see Figure 5).

> **The Effect of Data Size on Generalization**
>
> Increasing the dataset size alone without increasing its diversity by, for example, increasing the number of transitions without also increasing the number of training environments, does not lead to significant performance improvements on new environments for any of the offline learning algorithms.

### 4.6 GENERALIZATION TO NEW INSTRUCTIONS USING HUMAN DEMONSTRATIONS

Here we aim to assess a different type of generalization, namely generalization to unseen initial state and reward functions. Moreover, we also want to test if our conclusions hold in more challenging and realistic domains, hence we benchmark three algorithms, BC, CQL and BCQ, on a challenging benchmark, WebShop. Figure 7 shows the train and test scores (where average score = average reward $*10$) and the success rate (i.e., % of rollouts which achieve the maximum reward of 10). While the average score measures how closely an agent policy is able to follow the given instruction, the success rate determines how correct the actions taken are. In our experiments, BC obtains a higher score but a lower success rate than BCQ. This indicates that BCQ only clicks "buy" on products where it has

high confidence, while BC may click even when its confidence is lower, thus collecting additional points for partially correct selections but not having as high a success rate as BCQ.

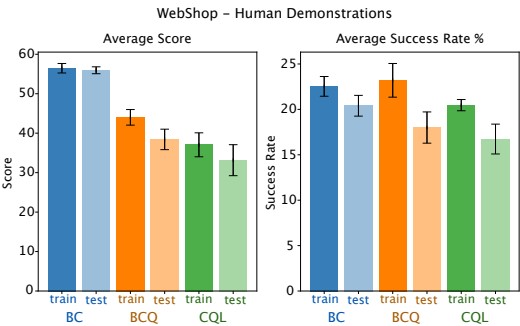

Figure 7: **Performance on WebShop.** Scores and success rates (%) for BC, BCQ, and CQL when trained on a dataset of human demonstrations from WebShop. Results were calculated by taking an average across 500 instructions (item descriptions) for both train and test. The mean and standard deviation were computed across 3 model seeds.

Following a similar evaluation procedure as in Yao et al. (2022), for each of our pre-trained models, we roll out the policy on the entire test set of goals (i.e. $\in [0, 500)$) and the first 500 goals from the train set (i.e $\in [1500, 2000)$). For each offline learning algorithm, we compute the mean and standard deviation of the average train and test scores and success rates using 3 model seeds.

As Figure 7 shows, on the human demonstration dataset, BC achieves a higher score than CQL and BCQ, on both train and test instructions from the human dataset. Note that the difference between train and test scores in BC is not very large. However, if trained for longer, all of these methods obtain much better training performance but their test performance starts decreasing, suggesting that they are prone to overfitting. During training, BCQ has a slightly higher success rate than BC and CQL. At test time, however, BC achieves the highest success rate, thus making BC a better-performing baseline in this domain. We report results on suboptimal demonstrations in Appendix L.1.

## 5 RELATED WORK

**Generalization in RL**    A large body of work has emphasized the challenges of training online RL agents that can generalize to new transition and reward functions (Rajeswaran et al., 2017; Machado et al., 2018; Justesen et al., 2018; Packer et al., 2019; Zhang et al., 2018a;b; Nichol et al., 2018; Cobbe et al., 2018; 2020; Juliani et al., 2019; Küttler et al., 2020; Grigsby & Qi, 2020; Chen, 2020; Bengio et al., 2020; Bertran et al., 2020; Ghosh et al., 2021; Kirk et al., 2023; Ajay et al., 2021; Ehrenberg et al., 2022; Lyle et al., 2022). A number of different RL environments have recently been created to support research on the generalization abilities of RL agents (Juliani et al., 2019; Küttler et al., 2020; Samvelyan et al., 2021; Frans & Isola, 2022; Albrecht et al., 2022). However, all of these simulators focus on benchmarking online rather than offline learning algorithms and don't have associated datasets. A natural way to alleviate overfitting is to apply widely-used regularization techniques such as implicit regularization (Song et al., 2020), dropout (Igl et al., 2019), batch normalization (Farebrother et al., 2018), or data augmentation (Ye et al., 2020; Lee et al., 2020; Laskin et al., 2020; Raileanu et al., 2021; Wang et al., 2020; Yarats et al., 2021; Hansen & Wang, 2021; Hansen et al., 2021; Ko & Ok, 2022). Another family of methods aims to learn better state representations via bisimulation metrics (Zhang et al., 2021a; 2020; Agarwal et al., 2021a), information bottlenecks (Igl et al., 2019; Fan & Li, 2022), attention mechanisms (Carvalho et al., 2021), contrastive learning (Mazoure et al., 2022), adversarial learning (Roy & Konidaris, 2020; Fu et al., 2021; Rahman & Xue, 2021), or decoupling representation learning from decision making (Stooke et al., 2021; Sonar et al., 2020). Other approaches use uncertainty-driven exploration (Jiang et al.), policy-value decoupling (Raileanu & Fergus, 2021), information-theoretic approaches (Chen, 2020; Mazoure et al., 2020), non-stationarity reduction (Igl et al.; Nikishin et al., 2022), curriculum learning (Jiang et al., 2021b; Team et al., 2021; Jiang et al., 2021a; Parker-Holder et al., 2022), planning (Anand et al., 2022), forward-backward representations (Touati & Ollivier, 2021), or diverse policies (Kumar et al., 2020b). Note that all these works consider the online rather than offline RL setting. More similar to our work, Mazoure et al. (2022) train an offline RL agent using contrastive learning based on generalized value functions, showing that it generalizes better than other baselines in some cases. However, they don't focus on creating offline RL benchmarks for generalization across both transition and reward functions, and they don't compare offline RL algorithms with other competitive approaches based on sequence modeling or transformer policies. Similarly Yang et al. (2023) studies the generalization capabilities of offline *goal-conditioned* RL (GCRL) methods and observes that existing GCRL methods do not generalize to unseen goals.

**Offline RL Benchmarks**  For many real-world applications such as education, healthcare, autonomous driving, or robotic manipulation, learning from offline datasets is essential due to safety concerns and time constraints (Dasari et al., 2019; Cabi et al., 2019; Li et al., 2010; Strehl et al., 2010; Thomas et al., 2017; Henderson et al., 2008; Pietquin et al., 2011; Jaques et al., 2020). Recently, there has been a growing interest in developing better offline RL methods (Levine et al., 2020; Prudencio et al., 2023; Kostrikov et al., 2021; Fujimoto et al., 2018; 2019; Agarwal et al., 2020; Nair et al., 2020; Fujimoto & Gu, 2021; Liu et al., 2020b; Zanette et al., 2021; Lambert et al., 2022; Rashidinejad et al., 2021; Yarats et al., 2022; Brandfonbrener et al., 2022; Wu et al., 2019; Zhang et al., 2021b; Cheng et al., 2022) which aim to learn offline from fixed datasets without online interactions with the environment. With it, a number of offline RL datasets have been created (Fu et al., 2020; Qin et al., 2022; Gulcehre et al., 2020; Brant & Stanley, 2019; Zhou et al., 2022; Ramos et al., 2021; Kurin et al., 2017). However, all these datasets contain trajectories collected from a single environment instance. In contrast, our collected datasets aim to evaluate an agent's ability to generalize to environment instances after being trained purely offline on a dataset of trajectories from similar yet distinct environment instances. V-D4RL (Lu et al., 2023) also proposes a benchmark suite for assessing the robustness of offline RL methods but in the continuous control domain. A number of large-scale datasets of human replays have also been released for StarCraft (Vinyals et al., 2017), Dota (Berner et al., 2019), MineRL (Guss et al., 2019), and MineDojo (Fan et al., 2022). However, training models on these datasets requires massive computational resources, which makes them unfeasible for academic or independent researchers. More similar to ours, (Hambro et al., 2022) introduces a large-scale offline RL dataset of trajectories from the popular game of NetHack (Küttler et al., 2020) consisting of 3 billion state-action-score transitions from 100,000 bot trajectories. In NetHack, each instance of the game is procedurally generated, meaning that the agent needs to generalize in order to perform well both during training and at test time. Here too, BC outperforms offline RL, which is in line with our results. However, this dataset requires significant resources to train highly performant agents, only considers generalization across different transition functions and not across different reward functions, and does not allow for a clear split between train and test scenarios. In contrast, one can train competitive agents on our datasets in just a few hours, making it a more accessible benchmark for generalization in offline RL that should enable fast iteration on research ideas.

## 6  CONCLUSION

In this paper, we compare the generalization abilities of online and offline learning approaches. Our experiments show that existing offline learning algorithms (including state-of-the-art offline RL, behavioral cloning, and sequence modeling approaches) perform significantly worse on new environments than online RL methods (like PPO). Our paper is first to introduce a benchmark for evaluating the generalization of offline learning algorithms. The absence of such a benchmark has historically limited our understanding of these algorithms' real-world applicability, so our work strives to bridge this gap. To achieve this, we release a collection of offline learning datasets containing trajectories from Procgen and WebShop.

Our results suggest that existing offline learning algorithms developed without generalization in mind are not enough to tackle these challenges, which is crucial in order to make them feasible for real-world applications. We observe that increasing dataset diversity can lead to significant improvements in generalization even without increasing the size of the dataset. Contrary to prior work on offline RL in singleton environments, we find that their generalization does not significantly improve with the size of the dataset without also enhancing its diversity. Hence, we believe more work is needed to develop offline learning algorithms that can perform well on new scenarios at test time.

## ACKNOWLEDGEMENTS

We would like to thank Sharath Raparthy for helping us in open-sourcing the code. We would also like to express our sincere gratitude to (in alphabetical order) Amy Zhang, Eric Hambro, and Mikayel Semvelyan for providing valuable feedback on this project.

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

# A    APPENDIX

We have open-sourced our codebase and datasets on GitHub[*]. The code repository consists of two separate folders: 1. `Procgen` and 2. `WebShop`. Each of these sub-repositories have a well-documented `README.md` with all the necessary steps needed to reproduce the experimental results in this paper to implement and train other offline learning models, or to generate other datasets and use them.

**License**    We have released the codebase and the datasets under a `CC-by-NC` license.

# B    LIMITATIONS AND FUTURE WORK

In this section, we discuss some potential limitations of our work and suggest future research avenues. First, our main datasets in Procgen have 1 million transitions. Contrasting this with many offline RL datasets like Atari (Agarwal et al., 2020) where the dataset consists of the entire training trajectory from the behaviour policy, the dataset size could have been inflated to 25M (since in Procgen, PPO uses 25M evironment steps to reach convergence). We believe this is a good step. However, training offline learning methods on such large datasets requires multi-GPU parallelism and significant computational resources, especially when using sequence modelling approaches. Thus, we opted for a more lightweight dataset which can enable a broad community of researchers to make progress on these problems without requiring extensive computational resources. Future work could explore the possibility of leveraging larger datasets, while seeking more efficient computational strategies.

We also note that all of our datasets have a discrete action space. Lately, numerous offline RL algorithms (Hansen-Estruch et al., 2023; Garg et al., 2023), have been developed exclusively for continuous action spaces. Adapting their loss function to discrete action space is a non-trivial task, so we leave it to future work to address this conversion problem which should extend the applicability of these algorithms.

Our datasets from the WebShop environment did not explore cross-product (i.e. training on selected instructions having some categories of products and testing on never-seen-before categories) generalization, an aspect that could be a fascinating direction for subsequent studies. The ability of models to generalize across diverse products could prove extremely useful in web-navigation environments. It is also worth considering the latest improvements like (Chen et al., 2023; Mohtashami & Jaggi, 2023), that make transformer models better at handling longer context length in situations like WebShop. This could help deal with the problem of long inputs in the state space. This might allow us to use sequential decision-making like the Decision Transformers that require multiple steps of observations in one context.

Lastly, Offline RL algorithms have a quadratic error bound on the horizon since there is no control over the data-generating policy and it may require evaluating out-of-distribution states (Levine et al., 2020). However, this theoretical observation has only been empirically validated on singleton environments. The goal of our study is to empirically evaluate offline RL algorithms on new environments rather than provide a theoretical explanation. Our results are in line with the theoretical results demonstrating that offline RL algorithms struggle with generalization but only in theory but also in practice. We hope this will inspire future work that aims to overcome these limitations, as well as a theory of why existing offline RL algorithms struggle to generalize to new environments (which better models our setting and would be an extension of prior work on this topic).

One promising avenue for future research is to combine offline RL methods with techniques that improve generalization in online RL such as data augmentation (Laskin et al., 2020; Raileanu et al., 2021; Yarats et al., 2021), regularization (Song et al., 2020; Igl et al., 2019; Farebrother et al., 2018), representation learning (Zhang et al., 2021a; 2020; Agarwal et al., 2021a; Mazoure et al., 2022), or other approaches focused on data collection, sampling, or optimization (Jiang et al., 2021b; Raileanu & Fergus, 2021; Jiang et al.). It is also possible that entirely new approaches that take into account the particularities of this problem setting will need to be developed in order to tackle the challenge of generalization in offline learning. As mentioned in the paper, sequence modeling approaches that rely on transformer-based policies cannot be directly applied to more complex environments

---

[*]`https://www.github.com/facebookresearch/gen_dgrl`

like WebShop due to their limited context lengths. Thus, we expect these methods will benefit from future advances in transformer architectures that can handle longer inputs. We hope our study and benchmark can enable faster iteration on research ideas towards developing more general agents that can learn robust behaviors from offline datasets.

## C  BROADER IMPACT

This paper proposes datasets for evaluating generalization of behaviors learned offline via behavioral cloning, offline reinforcement learning, and other approaches. We also evaluate state-of-the-art methods on this benchmark, concluding that more work is needed to train agents that generalize to new scenarios after learning solely from offline data. On the one hand, improving the generalization of offline learning methods can be important for developing more robust and reliable autonomous agents that take reasonable actions even in states that they haven't encountered during training, which are likely to be common once such agents are deployed in real-world applications. On the other hand, deploying autonomous agents for high-stakes applications can have negative consequences, so additional safety precautions should be taken when considering deployment in the real-world. Since our results are based on simulated environments for video games and e-commerce like Procgen and WebShop, which are somewhat simplified compared to real-world settings, we do not foresee any direct negative impact on society.

## D  EXTENDED RELATED WORK

**Multi-task RL**   Recently, significant attention has been directed towards the large-scale multi-task offline RL and meta RL settings. Notably, pioneering multi-task approaches such as those outlined in Kumar et al. (2022); Lee et al. (2022); Taiga et al. (2022) have showcased their efficacy in harnessing extensive multi-task datasets based on Atari Agarwal et al. (2020). However, these settings are different from the problem we consider our this paper. To elaborate, Lee et al. (2022) pre-trains the model on select Atari games, followed by subsequent fine-tuning on the remaining ones. Similarly, a lot of meta RL works (Li et al., 2021; Pong et al.; Zhao et al., 2021; Dorfman et al., 2021; Mitchell et al., 2020) also try to tackle this problem but require some finetuning at test time. In contrast, our benchmark centers around the evaluation of zero-shot generalization which is a more challenging setting with wider applicability in the real-world. Similarly, the works presented in Kumar et al. (2020a) and Taiga et al. (2022) delve into the realm of inter-game generalization in Atari, employing a training dataset comprising of approximately 40 games. Our focus, in contrast, resides in scrutinizing intra-game inter-level generalization within the Procgen framework. It is worth highlighting that our dataset architecture prioritizes memory efficiency, allowing for seamless execution by the academic community without necessitating access to a large number of GPUs, a requirement which, regrettably, constrained the feasibility of the approaches proposed in (Kumar et al., 2020a) and (Taiga et al., 2022) that heavily relies on many TPUs. This emphasis on accessible resources aligns with our intent to facilitate broader engagement and reproducibility in this research area.

## E  DISCUSSION

In this section, we discuss a few points that we believe are the reasons why BC generalizes better than offline RL, even in the presence of suboptimal data. We believe the reason offline RL methods fall behind BC is that they adopt a risk-averse approach, avoiding actions not encountered during training. This becomes a limitation when agents are tested in new environments, as they are likely to default to suboptimal policies due to unfamiliar states. On the other hand, BC, unbounded by these constraints, utilizes its learned representations to select the best action by identifying the most similar training state to the current test state. If BC effectively learns state representations, it could generalize well in new environments. Regarding why offline learning methods are outperformed by online RL, we think that the advantage of online RL comes from its ability to gather and learn from a broader range of states through its own data collection, as opposed to the fixed dataset in BC and offline RL (Agarwal et al., 2020). This exposure to a variety of states enables better learning of representations and decision-making in new scenarios. Our experiments in Section 4.4 demonstrate that training with more varied data significantly enhances offline learning methods' generalization.

However, as indicated in Section 4.2, merely using data from multiple PPO checkpoints (i.e. in the case of suboptimal dataset) is not sufficient. This data, being sparsely sampled, doesn't cover the entire state space. Understanding how training dynamics affect data diversity is an area worth exploring in future research.

# F  DATASET DETAILS

## F.1  PROCGEN

**Environment** Procgen (Cobbe et al., 2020) is an online RL benchmark that serves to assess generalization in 16 different 2D video games. Procgen makes use of procedural content generation in order to generate a new level (corresponding to a particular seed) when the episode is reset. This way an unlimited number of varying levels can be generated for each game, each level having different dynamics, layouts, and visual appearances (such as background colors and patterns, or number and location of various items and moving entities), but the same reward function. This makes Procgen a good benchmark for testing an agent's ability to generalize to *new environment instances (levels) with unseen initial states and transition functions but the same reward function*.

A single transition in Procgen comprises of an observation (represented by an RGB image of shape 64x64x3), a discrete action (the maximum action space is 15), a scalar reward (which can be dense or sparse depending on the game), and a boolean value indicating whether the episode has ended.

**Offline Data Collection** Each level of a Procgen game is procedurally generated by specifying the $level\_seed$ which is a non-negative integer. We use levels $[0, 200)$ for collecting trajectories and offline training, levels $[200, 250)$ for hyperparameter tuning and model selection, and levels $[250, \infty)$ for online evaluation of the agent's performance.

To generate the offline learning datasets based on Procgen, for each game, we train 3 PPO (Schulman et al., 2015) policies (with random seeds 0, 1, and 2) using the best hyperparameters found in (Raileanu & Fergus, 2021) for 25M steps on the easy version of the game. We save model checkpoints once every 50 epochs (from a total of 1525 epochs). We then use these checkpoints to collect trajectories for constructing an *expert dataset* and a *suboptimal dataset*.

To create the *expert dataset*, we roll out the final checkpoint from a single pretrained PPO model, also referred to as the expert policy, in the training levels (i.e., allow it to interact online with the environment) and store 1 million $\{state, action, reward, terminated\}$ transitions for each Procgen game. Figures 24 and 25 show the number of episodes and transitions per level respectively. As can be seen, there can be variation in the number of transitions across the 16 games since some games have shorter episodes than others.

To create the *mixed expert-suboptimal datasets*, we evenly combine data from both the expert PPO checkpoint and another checkpoint that achieves 50% of the expert's performance. Consequently, these datasets showcase average episodic returns that are approximately 75% of those seen in the expert datasets. Table 1 shows the average return per dataset collected by different PPO model seeds for 1M transition steps. Since we ran our experiments on the 1M expert dataset collected from Seed 1, we also collected this suboptimal dataset using PPO's checkpoints from this model seed only. Figures 26 and 27 show the number of episodes per level and total number of transitions per level for each game in Procgen from the 1M *mixed expert-suboptimal* dataset (seed 1) which was used for all experiments in Section 4.2.

**Data Storage** For each episode, we store all the corresponding $\{state, action, reward, terminal\}$ transitions. Each trajectory is then stored as a single $.npz$ file, with the name of $timestamp\_index\_length\_level\_return$. This naming convention allows for analyzing and working with trajectories filtered by level or return.

## F.2  WEBSHOP

**Environment** WebShop (Yao et al., 2022) is a text-based web-navigation environment, built to assess the natural language instruction following and sequential decision making capabilities of language-based agents. In this environment, there are two types of tasks: search and choice. For the

Table 1: **Procgen 1M Dataset Average Return:** This table shows the average return of each game's dataset collected by different model seeds of the expert PPO policy, as well as the suboptimal dataset.

| Game | Mixed Expert-Suboptimal | Expert | | | |
|---|---|---|---|---|---|
| | Seed 1 | Mixed | Seed 0 | Seed 1 | Seed 2 |
| Bigfish | 10.61 | 9.57 | 12.03 | 14.01 | 6.69 |
| Bossfight | 6.05 | 8.31 | 8.32 | 8.10 | 8.62 |
| Caveflyer | 4.95 | 7.23 | 6.74 | 6.97 | 7.79 |
| Chaser | 5.32 | 6.50 | 5.95 | 6.68 | 6.93 |
| Climber | 6.37 | 8.58 | 8.30 | 8.53 | 8.59 |
| Coinrun | 7.16 | 9.42 | 9.23 | 9.64 | 9.33 |
| Dodgeball | 3.50 | 5.46 | 5.97 | 4.61 | 6.26 |
| Fruitbot | 22.36 | 29.21 | 29.97 | 29.81 | 29.15 |
| Heist | 5.96 | 7.85 | 7.84 | 7.97 | 7.26 |
| Jumper | 6.88 | 8.55 | 8.53 | 8.47 | 8.50 |
| Leaper | 2.03 | 2.68 | 2.68 | 2.71 | 2.72 |
| Maze | 6.83 | 9.35 | 9.44 | 9.22 | 9.33 |
| Miner | 9.40 | 12.68 | 12.78 | 12.46 | 12.74 |
| Ninja | 6.01 | 8.07 | 7.85 | 8.05 | 8.00 |
| Plunder | 4.15 | 5.26 | 5.34 | 5.53 | 5.05 |
| Starpilot | 20.93 | 27.23 | 26.75 | 27.69 | 29.34 |

search task, the agent has to learn to generate relevant keywords based on a description of the desired product in order to increase the likelihood of getting a good product match within the search results. For the choice task, the agent needs to look through the search page and each item's individual page to select, customize, and buy the item that matches all the attributes mentioned in the instruction. Since the scope of our study is to assess the sequential decision making capabilities of offline agents, we limit our study to only the choice task. For the search task, we use a pre-trained BART model (Lewis et al., 2020) used in (Yao et al., 2022) to generate a search query at test time and then continue rolling out our pre-trained policies.

**Offline Data Collection**   For WebShop, we collect two types of offline learning datasets based on: (i) the *human demonstration dataset* provided by the authors which allows us to create a fixed size dataset of high quality (since the human demonstrations can be considered a gold standard), and (ii) the *imitation learning (IL) policy* pre-trained on these human demonstrations which allows us to create multiple datasets of varying sizes in order to study how performance scales with the dataset size and diversity. We also collect environment rewards for 452 out of 1571 human demonstrations provided by simulating the trajectories via the gym environment provided in WebShop's source code[*]. The initial state of this environment is determined by a random selection of 10 English letters. So we call the reset function repeatedly until the environment generates an instruction which is in the dataset. We then execute the actions of the episode from the dataset, and verify that the states returned by the environment per each step are the same as those in the dataset (except the last state, as the dataset doesn't store the *confirmation* state once an episode is completed). The rewards were collected on the fly and we stored them under the key "rewards" together with "states", "actions", etc. This way, we were able to collect 452 trajectories with their corresponding per-step rewards (out of 1571 in the human dataset) from the WebShop environment. Following the original paper, we represent observations using only the text modality. In the human dataset, we have 452 episodes, wherein the train split has 398 episodes,  3.7k transitions and an average reward of 7.54, and the evaluation split has has 54 episodes, 406 transitions and an average reward of  8. We also use the final IL checkpoint provided by the authors of WebShop to collect datasets of different sizes, i.e. $\in 100, 1000, 1500, 5000, 10000$ episodes, where in all of these datasets the average reward is  5.8-5.9. In this case, a larger dataset also has a greater diversity of environment instances specified by different natural language instructions (or item descriptions).

**Data Storage**   Following Yao et al. (2022), we store all episodes in a single $.json$ file.

---
[*] https://github.com/princeton-nlp/WebShop

# G EXPERIMENTAL SETUP

We collect data from 200 different Procgen levels for offline training, validate the hyperparameters online to perform model selection on the another 50 levels, and evaluate the agents' online performance on the remaining levels, i.e. level_seed $\in [250, \infty)$. For more details on the process for collecting these datasets and their reward distributions, see Appendix F. We also explain the architecture used for the underlying policy in Appendix H.2.1.

## G.1 BASELINES

1. **Behavioral Cloning (BC)** is trained to predict the actions corresponding to all states in the dataset, via cross-entropy loss. This baseline is parameterized by either a ResNet (He et al., 2016) (in the case of Procgen) or a BERT (Devlin et al., 2019) (in the case of WebShop), takes as input the current state and outputs a probability distribution over all possible actions.

2. **Batch Constrained Q-Learning (BCQ) Fujimoto et al. (2019)** restricts the agent's action space to actions that appear in the dataset for a given state, in an effort to reduce distributional drift which is one of the main challenges in offline RL.

3. **Conservative Q-Learning (CQL) Kumar et al. (2020a)** regularizes the Q-values by adding an auxiliary loss to the standard Bellman error objective, in an effort to alleviate the common problem of value overestimation in off-policy RL.

4. **Implicit Q-Learning (IQL) Kostrikov et al. (2021):** uses expectile regression to estimate the value of the best action in a given state, in order to prevent evaluating out-of-distribution actions.

5. **Behavioral Cloning Transformer (BCT) Chen et al. (2021)** is a transformer-based version of BC, where the agent's policy is parameterized by a causal transformer with a context containing all previous (state, action) pairs in the episode. The agent has to predict the next action given the current state and episode history.

6. **Decision Transformer (DT) Chen et al. (2021)** is similar to BCT but in addition to the state and action tokens, the context also contains the return-to-go at each step. The agent has to predict the action and the current return-to-go given state and episode history. At test time, DT is conditioned on the maximum possible return-to-go for that particular game.

All our implementations have almost 1 million parameters with ResNet He et al. (2016) encoder and were trained and evaluated in a similar manner as detailed in Appendix H.2.1.

## G.2 EVALUATION METRICS

For Procgen, we report the mean and standard deviation across either 3 or 5 model seeds for each game, as well as the inter-quartile mean (IQM) (Agarwal et al., 2021b) and mean normalized return averaged across all 16 games. We follow Agarwal et al. (2020) which showed that IQM is a more robust metric than the mean or median when reporting aggregate performance across multiple tasks, especially for a small number of runs per task. For WebShop, we follow the recommended procedure in Yao et al. (2022) and report the average scores and success rates on a set of train and test instructions.

# H HYPERPARAMETERS

## H.1 BEHAVIOUR POLICY

**Procgen** To collect datasets for each of the 16 games within Procgen, we employed the Proximal Policy Optimization (PPO) algorithm (Schulman et al., 2017) using the setup outlined in Raileanu & Fergus (2021). Specifically, our PPO training involved training the policy for 25 million environment steps, utilizing a set of 200 training levels. The architecture consists of a ResNet (He et al., 2016) which encodes the 64x64 RGB images into a linear embedding, which is then processed by two parallel fully-connected layers, one for the actor and one for the critic with hidden dimension of 256. The policy is trained for 25M environment steps and checkpoints are saved regularly throughout

this process. All of the hyperparameters are same as the one used in (Raileanu & Fergus, 2021) and (Cobbe et al., 2020) and are detailed in Table 2 which were shared across all 16 games.

Table 2: Table summarizing the hyperparameters used for PPO in Procgen

| Hyperparameter | Value |
|---|---|
| $\gamma$ | 0.999 |
| $\lambda_{\text{GAE}}$ | 0.95 |
| PPO rollout length | 256 |
| PPO epochs | 3 |
| PPO minibatches per epoch | 8 |
| PPO clip range | 0.2 |
| PPO number of workers | 1 |
| Number of envs per worker | 64 |
| Adam learning rate | 5e-4 |
| Adam $\epsilon$ | 1e-5 |
| PPO max gradient norm | 0.5 |
| PPO value clipping | no |
| return normalization | yes |
| value loss coefficient | 0.5 |
| entropy bonus | 0.01 |

To provide an overview of the policy's performance, Figure 8 depicts the training returns across each game for the entire 25 million environment steps for all 3 model seeds. Our expert dataset, which is obtained online, is derived from the final checkpoint of this training procedure. Furthermore, we curated a suboptimal dataset by selecting a checkpoint having performance at 50% of the expert's level. The achieved returns for these checkpoints in each game are listed in Table 1.

**WebShop**  We employed a pre-trained Imitation Learning (IL) checkpoint provided by (Yao et al., 2022) to collect training trajectories by rolling out the policy in the online environment. This particular checkpoint achieves a score of 59.9 (which corresponds to a mean reward of 5.99) and success rate 29.1% in the test set. This policy consists of a 110M parameter BERT model for encoding the current states and a list of available actions, and outputs log probabilities over the available actions.

## H.2   MODEL TRAINING

Here we list down the set of hyperparameters used in each offline learning algorithm separately. Moreover, all offline RL baselines (i.e. BCQ, CQL and IQL), plus BC, had the same encoder size and type, which was a ResNet in the case of Procgen, and a BERT encoder in the case of WebShop. All of our experiments were run on a single NVIDIA V100 32GB GPU on the internal cluster, with varying training times and memory requirements.

### H.2.1   PROCGEN

**Hyperparameters**  For BC, we performed a sweep over the batch size $\in \{64, 128, 256, 512\}$ transitions and learning rate $\in \{5e-3, 1e-4, 5e-4, 6e-5\}$. For BCQ, CQL, and IQL, which use a DQN-style training setup (i.e., they have a base model and a frozen target model) (Mnih et al., 2013), in addition to the hyperparameters mentioned for BC, we swept over whether to use polyak moving average or directly copy weights, in the latter case, the target model update frequency $\in \{1, 100, 1000\}$ and in the former case, the polyak moving average constant $\tau \in \{0.005, 0.5, 0.99\}$. For BCQ, we also swept over the threshold value for action selection $\in \{0.3, 0.5, 0.7\}$. For CQL, we swept over the CQL loss coefficient, which we refer to as `cql_alpha` in our codebase, $\in \{0.5, 1.0, 4.0, 8.0\}$. Finally, for IQL, we sweep over the temperature $\in \{3.0, 7.0, 10.0\}$ and the expectile weight $\in \{0.7, 0.8, 0.9\}$.

For sequence modelling algorithms, DT and BCT, we sweep over the learning rate and batch size mentioned above, as well as the context length size $\in \{5, 10, 30, 50\}$. For DT, we also sweep over the return-to-go (rtg) multiplier $\in \{1, 5\}$. We follow similar approach in (Chen et al., 2021) to set the maximum return-to-go at inference time by finding the maximum return in the training dataset for a

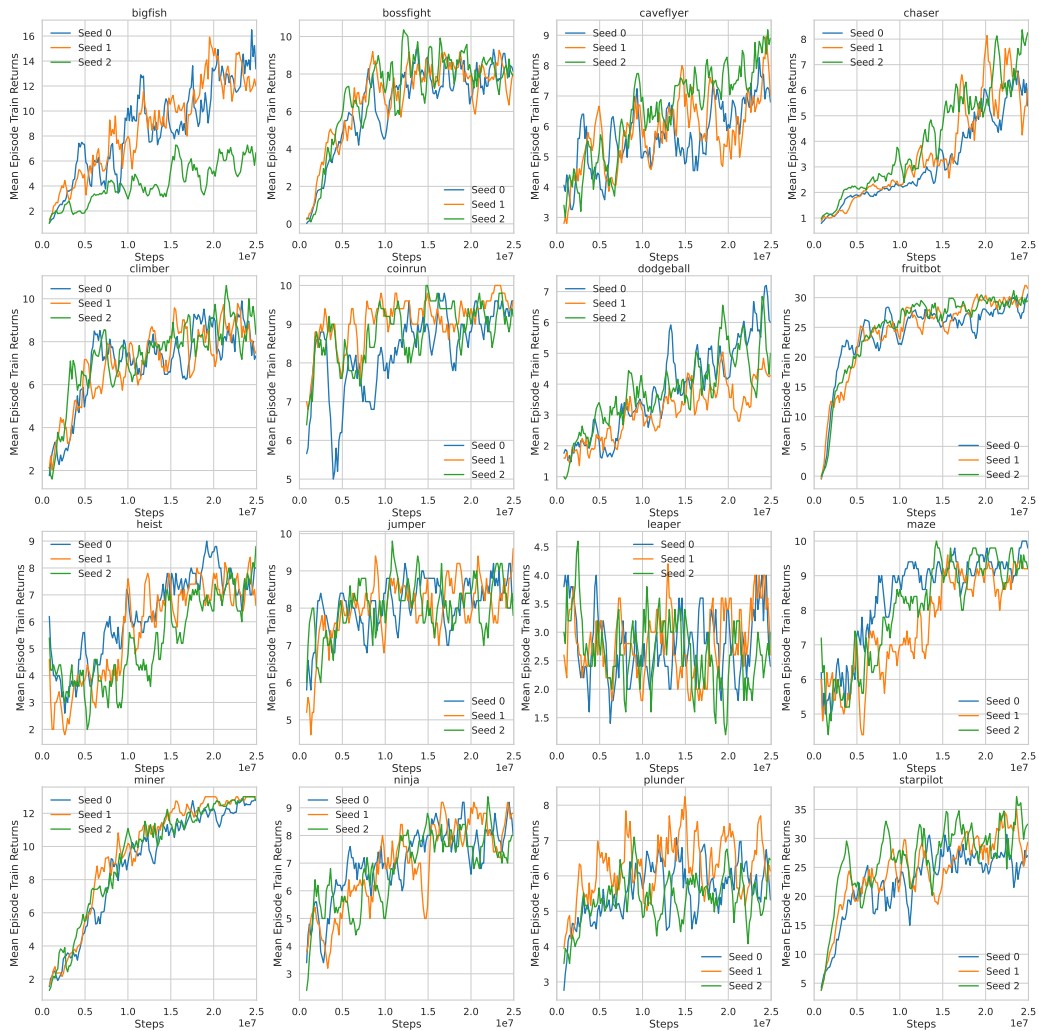

Figure 8: Training Returns for the Procgen data collecting behavioral policy PPO.

particular game and then multiplying by either 1 or 5 depending on the rtg multiplier value. We also use the default value of 0.1 for dropout in DT and BCT as was used originally in (Chen et al., 2021) for optimal performance.

We run 3 random trials per each configuration and select the best hyperparameter by looking at the min-max normalized mean train and validation results, averaged across all 16 games. Train results are calculated by rolling out the final checkpoint of the policy 100 times on training level and likewise for validation results, the policy is rolled out 100 times by randomly sampling one level at a time out of 50 validation levels. During our initial experiments we noticed that many of these algorithms overfitted quickly (within 10-20 epochs) on the training dataset. Therefore, to save training time and prevent overfitting, we employ early stopping by calculating the validation return after every epoch and stopping the training process if the validation return does not improve in the last 10 epochs. However, in Section 4.5, where the dataset size was either 5M or 10M transitions, we used fixed 3 epochs only.

Table 3 list the final hyperparameters for BC, BCQ, BCT, DT, CQL and IQL for 1M expert and suboptimal dataset, 10M expert dataset as well as for the single level experiments (except for IQL, which uses polyak averaging in single-level experiments). Since the 25M suboptimal dataset, which was used in Section 4.2, has a very different distribution than our synthetically created 1M mixed expert-suboptimal dataset (which had 75% returns of experts'), we ran a hyperparameter sweep on this dataset by uniformly sampling 1M transitions and following a similar procedure as above. We

| Algo | Hyperparameter | 1M Expert | 1M Mixed Suboptimal | 25M Suboptimal | 10M Expert |
|------|----------------|-----------|---------------------|----------------|------------|
| **BC** | Learning Rate | 0.0005 | 0.0005 | 0.0005 | 0.0005 |
| | Batch Size | 256 | 256 | 256 | 256 |
| **BCT** | Learning Rate | 0.0005 | 0.0005 | 0.0005 | 0.0005 |
| | Batch Size | 512 | 512 | 512 | 512 |
| | Context Length | 30 | 5 | 5 | 5 |
| | Eval Return Multiplier | 0 | 0 | 0 | 0 |
| **DT** | Learning Rate | 0.0005 | 0.0005 | 0.0005 | 0.0005 |
| | Batch Size | 512 | 512 | 512 | 512 |
| | Context Length | 10 | 5 | 5 | 5 |
| | Eval Return Multiplier | 5 | 5 | 5 | 5 |
| **BCQ** | Learning Rate | 0.0005 | 0.0005 | 0.0005 | 0.0005 |
| | Batch Size | 256 | 256 | 512 | 256 |
| | Target model Weight Update | Direct copy | Polyak | Direct copy | Direct copy |
| | $\tau$ | - | 0.5 | - | - |
| | Target update frequency | 1000 | 1000 | 1000 | 1000 |
| | Threshold | 0.5 | 0.5 | 0.5 | 0.5 |
| **CQL** | Learning Rate | 0.0005 | 0.0005 | 0.0005 | 0.0005 |
| | Batch Size | 256 | 256 | 256 | 256 |
| | Target model Weight Update | Direct copy | Polyak | Direct copy | Direct copy |
| | $\tau$ | - | 0.99 | - | - |
| | Target update frequency | 1000 | 1000 | 1000 | 1000 |
| | Alpha | 4.0 | 4.0 | 4.0 | 4.0 |
| **IQL** | Learning Rate | 0.0005 | 0.0005 | 0.0005 | 0.0005 |
| | Target model Weight Update | Direct copy | Polyak | Direct copy | Direct copy |
| | Batch Size | 512 | 256 | 512 | 512 |
| | $\tau$ | - | 0.005 | - | - |
| | Target update frequency | 100 | 100 | 1000 | 100 |
| | Temperature | 3.0 | 3.0 | 3.0 | 3.0 |
| | Expectile | 0.8 | 0.8 | 0.8 | 0.8 |

Table 3: List of hyperparameters used in Procgen experiments

performed a similar sweep for the 10M expert dataset experiment as well. The best hyperparameters for this dataset are also listed in Table 3.

### H.2.2 WEBSHOP

**Hyperparameters**   For BC, we performed a sweep over the batch size $\in \{1, 4, 8\}$ and learning rate $\in \{2e-5, 2e-4, 2e-3\}$. For BCQ and CQL, we swept over the target model update frequency $\in \{100, 1000\}$. For BCQ, we also swept over the threshold value for action selection $\in \{0.1, 0.5, 0.9\}$ and for CQL $\alpha$, we swept over $\{4.0, 7.0\}$.

For the scaling experiment in Section L.1, the sweep for BCQ threshold was even wider $\in \{0.01, 0.1, 0.2, 0.3, 0.4, 0.5, 0.6, 0.7, 0.8, 0.9\}$ with rest of the hyperparameters being the same from the best hyperparameter selected above.

We run 1 trial for each combination of hyperparameters and select the best performing one by rolling out the agent on all validation goal levels (`goal_idx=(500, 1500)`) and on the first 500 train goal levels (`goal_idx=(500, 1500)`). Final hyperparameters are listed in Tables 4 for BC, CQL and BCQ.

| Algorithm | Hyperparameter | Human Demonstrations | IL Trajectories |
|---|---|---|---|
| **BC** | Learning Rate | 0.00005 | 0.00005 |
| | Batch Size | 1 | 1 |
| **BCQ** | Learning Rate | 0.00005 | 0.00005 |
| | Batch Size | 1 | 1 |
| | $\tau$ | 0.005 | 0.005 |
| | Target update frequency | 100 | 1000 |
| | Threshold | 0.5 | 0.9 |
| **CQL** | Learning Rate | 0.00005 | 0.00005 |
| | Batch Size | 1 | 1 |
| | $\tau$ | 0.005 | 0.005 |
| | Target update frequency | 1000 | 100 |
| | Alpha | 7.0 | 4.0 |

Table 4: List of hyperparameters used in WebShop experiments

## I   TRAINING AND TESTING ON SINGLE LEVEL IN PROCGEN

Here we show the results when training agents on expert and suboptimal transitions from `level 1` and testing online only on that level. Figures 13c (overall IQM) and 11 (per-game returns) show the performance of all baselines after training and testing on expert dataset from level 1, demonstrating that not just BC, but other offline RL baselines can learn well when trained on high-return demonstrations from singleton environments. Note that offline RL methods still seem to struggle more on some of the games, even when trained on demonstrations from a single level. Also note that here even the final checkpoint of PPO struggles in some games (achieving 0 reward), and that is why we report performance on `level 40` as well in Figures 13b, 9 and 10. Figure 13d and Figure 12 show the performance of these baselines when trained on suboptimal dataset in `level 1`. Here, in 9 out of 16 games, offline RL performs comparable or even better (in Bigfish, Fruitbot, Heist, Jumper) compared to BC and sequence modelling.

Figure 9: Performance of each baseline on level 40 in each Procgen game when **trained and tested on the same level using expert dataset**.

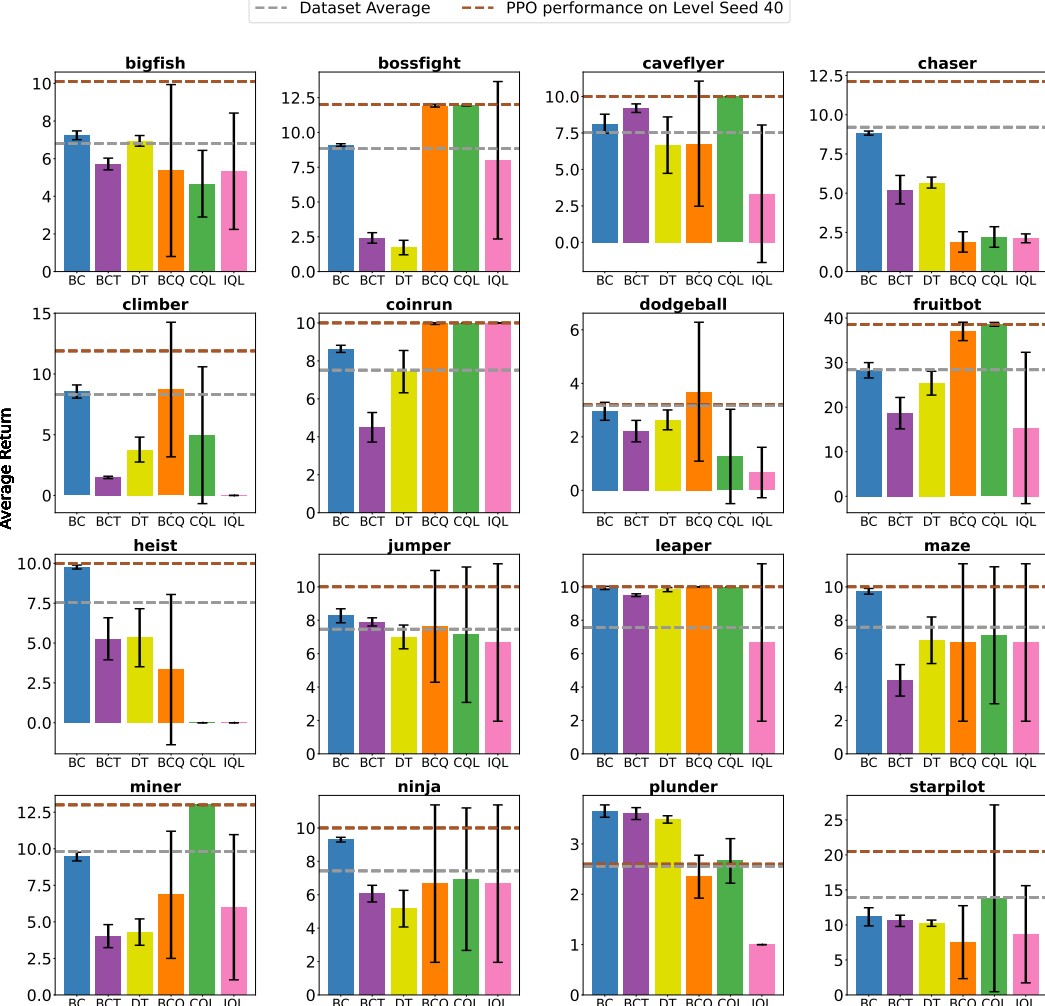

Figure 10: Performance of each baseline on level 40 in each Procgen game when **trained and tested on the same level using mixed expert-suboptimal dataset**.

## J    EFFECT OF DATA DIVERSITY ON GENERALIZATION: SUBOPTIMAL DATA

Here we use the entire training log of the behavioral policy (PPO) (Appendix H.1) and use a subset of interactions made by that agent as our offline dataset (see seed 1 in Figure 8) which has 25M transitions in total. *This approach of using the training trajectory of the behavioural policy and sampling mixed data out of it is also consistent with the previous literature ((Agarwal et al., 2020), (Gulcehre et al., 2020), (Kumar et al., 2020a), (Fu et al., 2020)).* We then uniformly sample episodes from the dataset such that we get a mixed, suboptimal-expert training dataset in Procgen and the dataset size has almost 1 million, 5 million as well as 10 million transitions. Throughout this scaling process, we keep all other hyperparameters same as well as the number of training levels constant at 200.

From Figures 14 and 18, it is evident that all the algorithms exhibit poor train and test performance, even with 10M transitions. In contrast with prior work showing that offline RL approaches learn well and even outperform BC when trained on suboptimal data, our experiments show that they still struggle when trained on data from multiple environments and tested on new ones, irrespective of the quality of the demonstrations (i.e., whether they are expert of suboptimal).

Figure 11: Performance of each baseline on level 1 in each Procgen game when **trained and tested on the same level using expert dataset**.

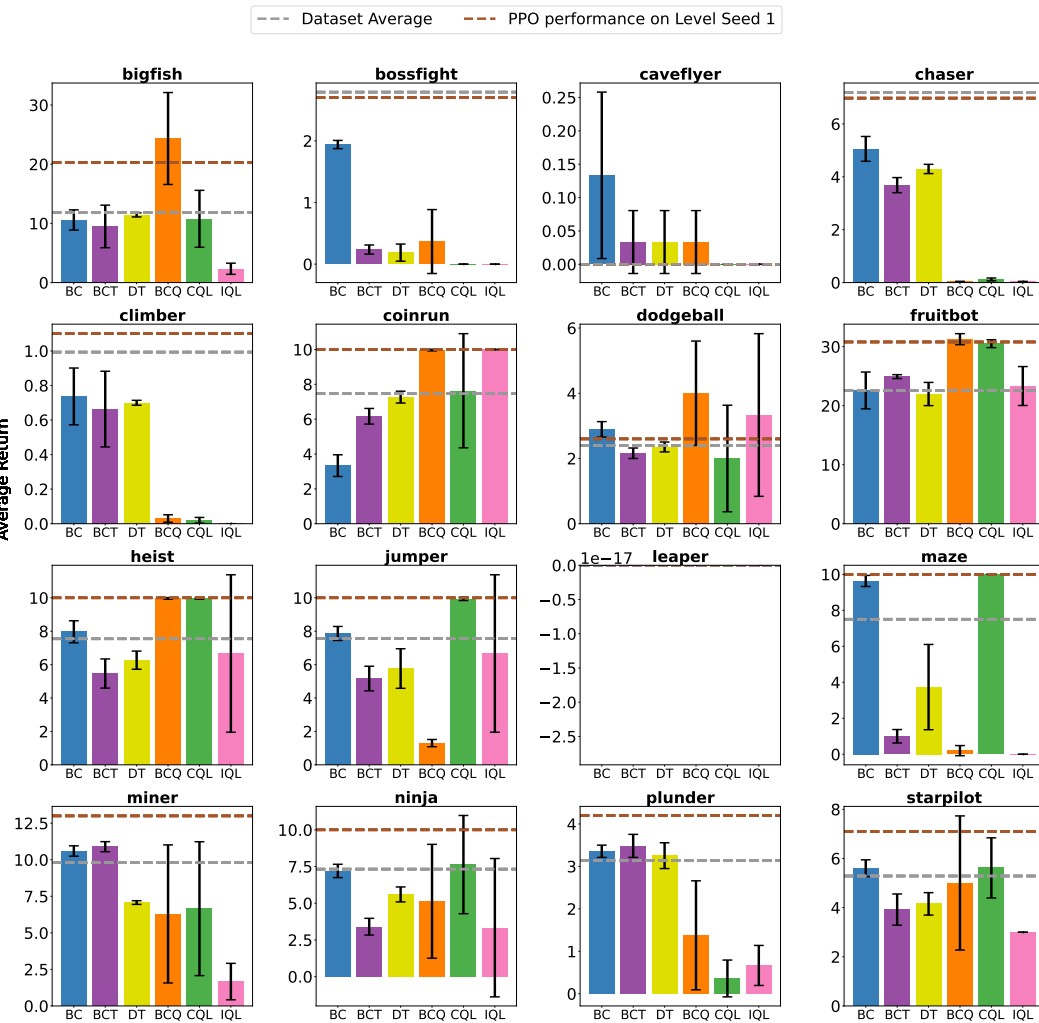

Figure 12: Performance of each baseline on level 1 in each Procgen game when **trained and tested on the same level using mixed expert-suboptimal dataset**.

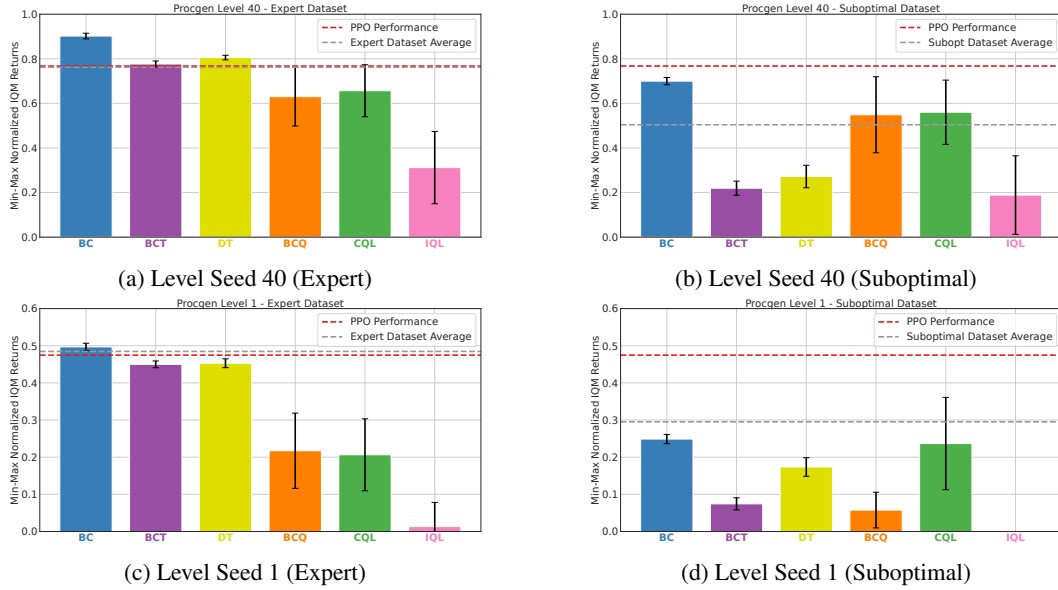

(a) Level Seed 40 (Expert)  (b) Level Seed 40 (Suboptimal)

(c) Level Seed 1 (Expert)  (d) Level Seed 1 (Suboptimal)

Figure 13: Aggregated performance of each baseline across all Procgen games when **trained and tested on the same level using suboptimal or expert dataset**. Red line represents the performance of our expert PPO checkpoint on this level.

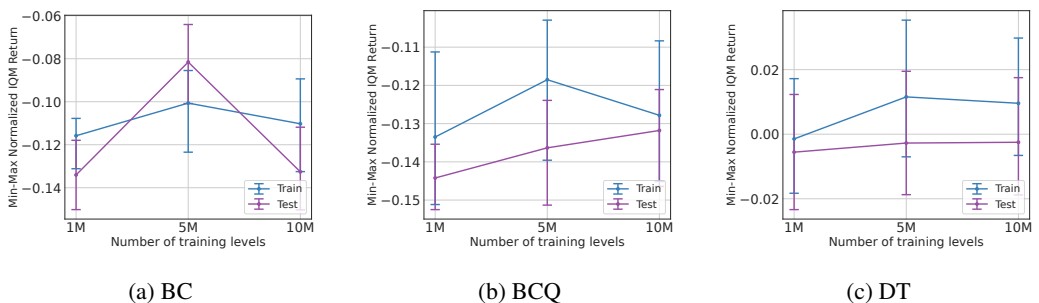

(a) BC  (b) BCQ  (c) DT

Figure 14: **The Effect of Data Size on Performance (Suboptimal)**. Train and test min-max normalized IQM scores for BC, BCQ, and DT as the size of the suboptimal dataset ia increased from 1 to 10 million transitions. All algorithms have poor train and test performance (even when using 10M transitions). While the train performance slightly increases with the dataset size, the test performance does not vary much. For remaining algorithms, see Figure 18.

## K    SCALING NUMBER OF VALIDATION LEVELS IN PROCGEN

In this section, we compare the use of a limited set of 50 validation levels in Section 5. We contrast this approach with a similar proportion of levels used for training. Specifically, if $[0, n)$ levels are allocated for training, then $[n, n + (0.1 * n))$ levels are reserved for validation. As illustrated in Figure 17, a notable trend emerges among all offline learning algorithms, indicating a robust correlation between the two aforementioned approaches. To maintain methodological consistency with our broader array of experiments presented in this paper, we advocate for the adoption of a consistent practice for this benchmark. Specifically, we suggest that researchers consider employing a level range of $[n, n + 50)$ exclusively for validation purposes.

## L    EFFECT OF DATA SIZE ON GENERALIZATION: MORE RESULTS

In this section, we perform the same experiment of scaling offline learning algorithms on the 10M expert dataset and 25M suboptimal dataset from Section 4.5 on the remaining offline learning algorithms: BCT, CQL and IQL. For both datasets, we observe similar findings, i.e. on expert dataset

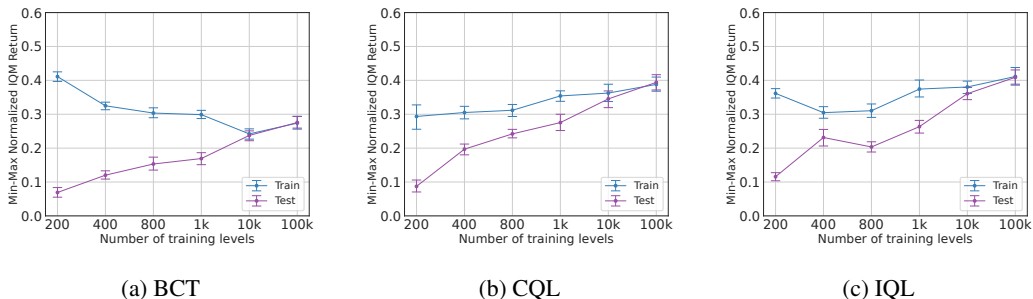

(a) BCT            (b) CQL            (c) IQL

Figure 15: **The Effect of Data Diversity on Performance of BCT, CQL & IQL.** Train and test performance for varying number of training levels in the 1M expert datasets, aggregated across all Procgen games. The plot shows the IQM and error bars represent the 75-th and 25th percentiles computed over 3 model seeds. While the training performance does not change much with the number of training levels, the test performance increases (and generalization gap decreases) with the diversity of the dataset.

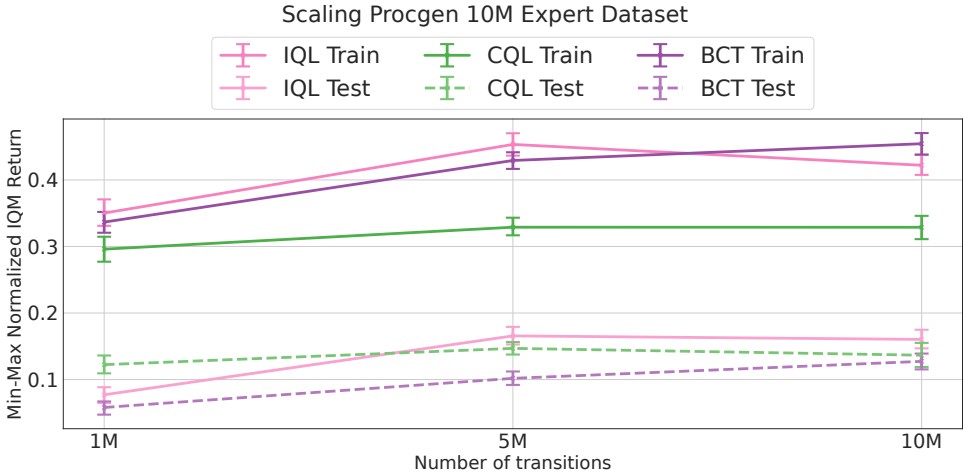

Figure 16: **The Effect of Data Size on Performance (Expert)**. Train and test min-max normalized IQM scores for BCT, CQL and IQL as the size of the dataset is increased from 1 to 10 million transitions aggregated across 5 random trials. Scaling the dataset size while keeping the number of training levels fixed (to 200) leads to a slight increase in both train and test returns, but the generalization gap remains about the same.

(Figure 16), all three methods have almost constant generalization gap as the data size scales up. On the suboptimal dataset (Figure 18), all methods have poor train and test aggregate performance and contrary to prior works, in our setup when we sample and train offline learning algorithms on a subset of episodes from the training log of the behavioral policy, the resulting offline learning policy does not generalize at all and does not even perform well on the 200 training levels as well.

## L.1 GENERALIZATION TO NEW INSTRUCTIONS USING SUBOPTIMAL DEMONSTRATIONS IN WEBSHOP

Here we train BC, CQL and BCQ on dataset collected using a pre-trained behavioral cloning policy (see Appendix F.2) to test the effect of scaling the number of episodes (and hence the variety of goal instructions) on the performance of these baselines. Figure 19 shows that in all three baselines, the train and test performance significantly increases once the dataset has at least 500 episodes. After that point, while there is not much increase in the scores in train and test levels, there is a slight increasing trend in the success rates for BCQ and CQL. *Overall, similar to Procgen, BC outperforms both BCQ and CQL on all datasets, thus highlighting the need for more research in this area*

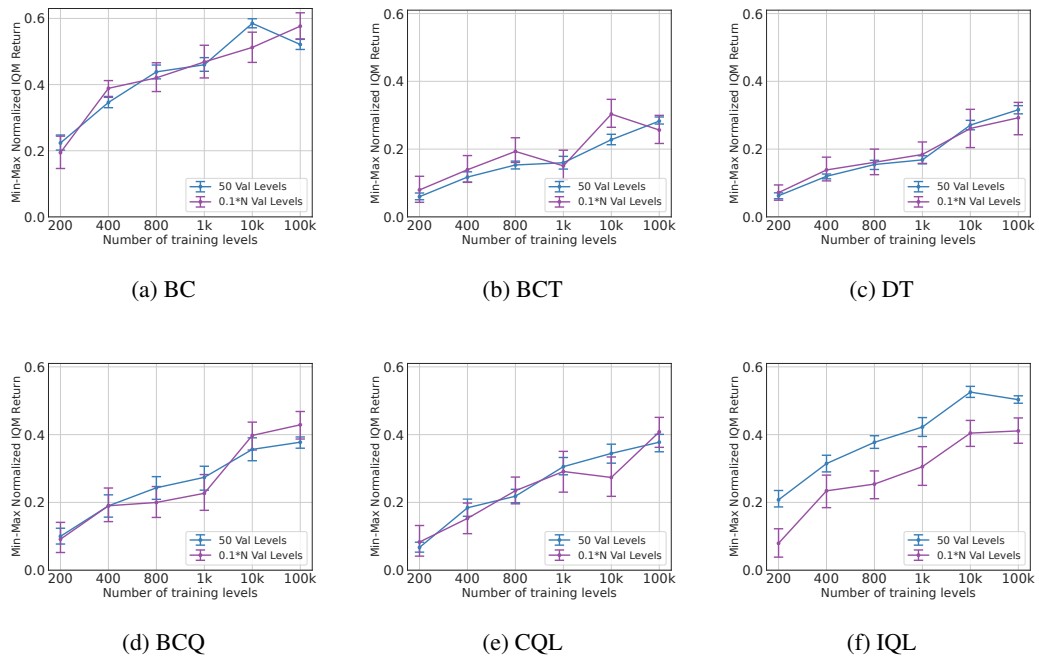

Figure 17: Comparing the use of 50 validation levels vs. validation levels proportional to the number of training levels when trained on 1M expert transitions with varying number of training levels in Procgen. IQM was computed using 3 model seeds for each experiment.

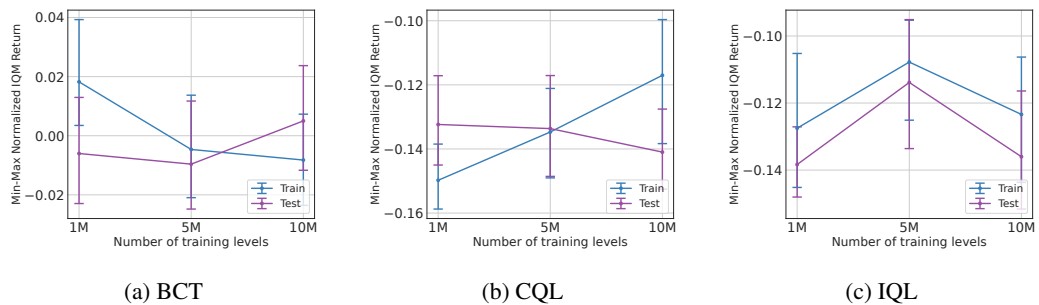

Figure 18: **The Effect of Data Size on Performance in Procgen (Suboptimal).** This plot shows the train and test min-max normalized IQM scores for BCT, CQL and IQL as the quantity of suboptimal offline dataset is increased from 1 million transitions to 10 million transitions. As can be seen, all algorithms have very poor train and test performance (even when using 10M transitions) and at a very granular level, the train performance generally increases, but test performance does not change much.

## M    MORE SCALING ANALYSIS

Here we try to see whether having fixed number of episodes per level (i.e. 10 episodes per level) as opposed to having fixed 1 million transitions in total irrespective of the number of levels leads to better generalization gains. For this, we collect many different datasets where we specify the total number of levels $\in [200, 400, 8001000]$ we want to collect the data from and on each level we collect 10 episodes (each episode has atleast 3 transitions). Hence, each dataset can have more than 1 million transitions in total.

We ran this experiment for a maximum of 1000 seeds (beyond which the dataset size became too huge to fit in a single GPU). In figure 20 we observe that in BC, BCQ, DT and BCT, the generalization gap decreases slightly, but the trend is similar to what we had observed in the case of fixed 1 million transitions in Section 4.4. Hence the amount of data per level in the underlying dataset does not affect the generalization performance of the offline RL method.

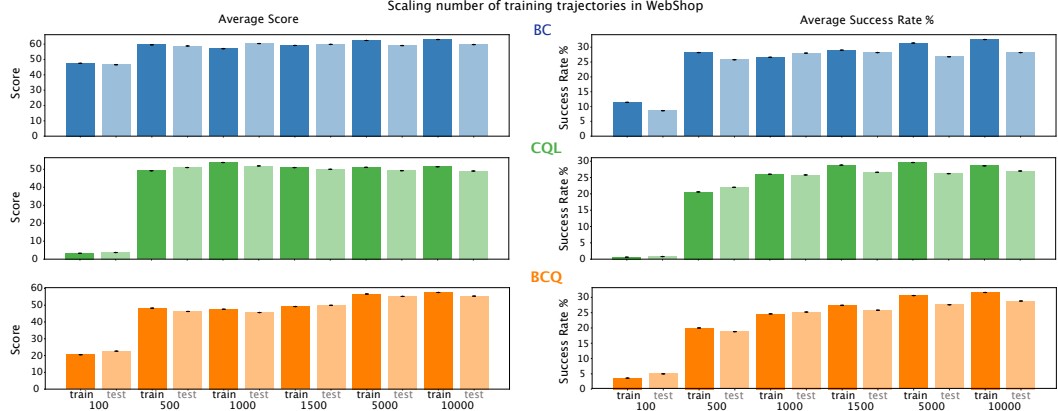

Figure 19: Score and success rate of each baseline on the WebShop environment when trained on various sizes of datasets. Once the dataset consists of 500 episodes, the performance increases significantly, however, beyond that the change in performance is not much.

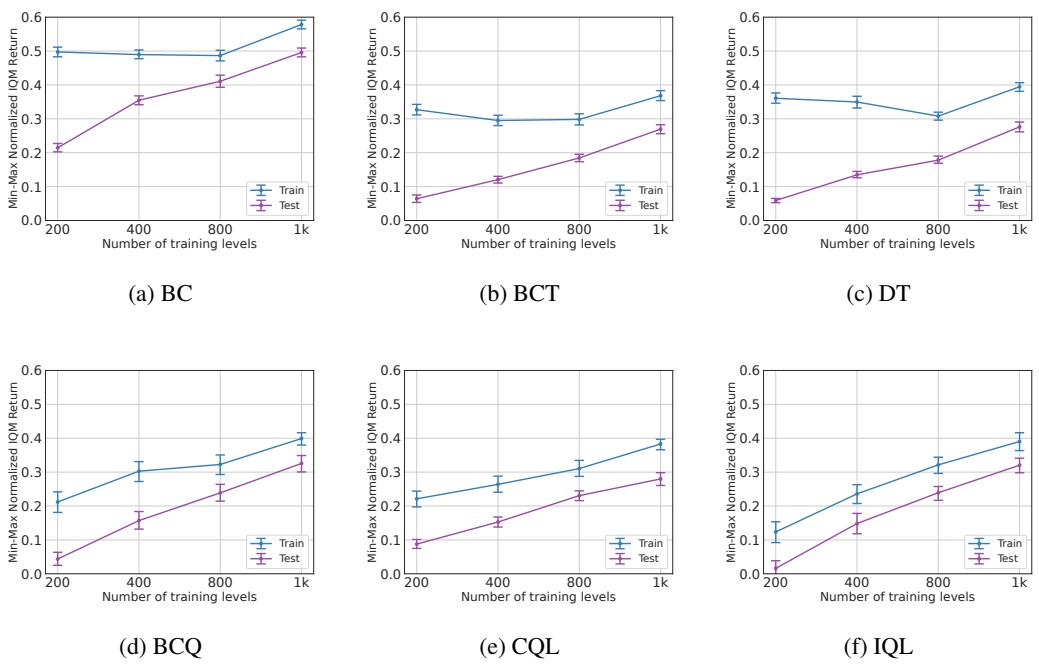

Figure 20: Scaling the training levels in the dataset while keeping the number of episodes per level fixed (10 episodes).

## N    PER-GAME SCORES IN PROCGEN

Figures 21 and 22 show the performance of these baselines on each individual game when trained using 1M expert dataset. In Figure 21, we compare the training returns which were calculated by averaging over 100 episodes by randomly sampling levels from the training set. The grey line shows the average return of the trajectories in the training dataset. For most games, at least some of these methods match the average performance of the training data. Among the different approaches, BC is the most robust, while the offline RL and sequence modeling approaches fail to do well on some of the games. In Figure 22, we compare the test returns which were calculated by averaging over 100 episodes by randomly sampling levels from the test set. The red line shows the average test performance of the final PPO checkpoint across 100 randomly sampled test levels.

Figure 21: **Procgen Per-Game Train Results on the 1M Expert Dataset.** Average episode return for each method on train levels for each Procgen game. The mean and standard deviation are computed across 5 model seeds. The grey line shows the average return of the trajectories in the training dataset. The brown line shows PPO's average train return which was computed over 100 randomly sampled train levels. Most of the methods match the average performance of the training data. In most cases, BC is competitive or better than the offline RL and sequence modeling approaches. For numerical comparison, refer to Table 5 in Appendix.

Figure 22: **Procgen Per-Game Test Results on the 1M Expert Dataset.** Average episode return for each method on test levels for each Procgen game. The mean and standard deviation are computed across 5 model seeds. The red line shows PPO's average test return which was computed over 100 randomly sampled test levels. As seen in previous figure, most of the methods match the average performance of the training data. However, many of them fail to reach PPO's performance at test time. In most cases, BC is competitive or better than the offline RL and sequence modeling approaches. For numerical comparison, refer to Table 5 in Appendix.

| Environment | BC | BCT | DT | BCQ | CQL | IQL |
|---|---|---|---|---|---|---|
| bigfish | **10.94 ± 0.75** | 9.99 ± 0.64 | 10.35 ± 0.69 | 6.23 ± 1.02 | 9.02 ± 1.37 | 8.19 ± 0.54 |
| bossfight | 7.16 ± 0.35 | 1.11 ± 0.15 | 1.01 ± 0.18 | 7.4 ± 0.36 | **8.57 ± 0.32** | 8.45 ± 0.36 |
| caveflyer | **6.84 ± 0.25** | 6.06 ± 0.15 | 6.56 ± 0.2 | 2.57 ± 0.39 | 2.65 ± 0.23 | 4.55 ± 0.31 |
| chaser | **5.6 ± 0.12** | 2.67 ± 0.23 | 2.98 ± 0.19 | 4.37 ± 0.56 | 3.94 ± 0.52 | 3.68 ± 0.13 |
| climber | 8.27 ± 0.22 | **8.35 ± 0.27** | 7.98 ± 0.27 | 2.19 ± 0.27 | 1.89 ± 0.17 | 5.22 ± 0.26 |
| coinrun | **9.48 ± 0.14** | 9.32 ± 0.06 | 9.42 ± 0.06 | 8.28 ± 0.18 | 8.58 ± 0.29 | 8.52 ± 0.26 |
| dodgeball | **4.01 ± 0.24** | 3.43 ± 0.06 | 3.9 ± 0.23 | 1.55 ± 0.15 | 1.7 ± 0.14 | 2.74 ± 0.25 |
| fruitbot | 27.35 ± 0.98 | 23.36 ± 0.34 | 23.5 ± 0.38 | 14.68 ± 1.29 | 19.42 ± 1.21 | **27.51 ± 0.51** |
| heist | **7.78 ± 0.26** | 7.0 ± 0.22 | 7.32 ± 0.12 | 3.96 ± 0.5 | 3.74 ± 0.31 | 5.02 ± 0.42 |
| jumper | 8.2 ± 0.06 | **8.74 ± 0.1** | 8.54 ± 0.13 | 7.14 ± 0.15 | 7.68 ± 0.16 | 7.8 ± 0.26 |
| leaper | 2.78 ± 0.18 | 2.34 ± 0.12 | 2.96 ± 0.2 | 2.54 ± 0.08 | 2.42 ± 0.23 | **3.08 ± 0.15** |
| maze | **9.0 ± 0.14** | 8.08 ± 0.2 | 8.74 ± 0.16 | 7.78 ± 0.22 | 7.34 ± 0.14 | 7.48 ± 0.13 |
| miner | **12.36 ± 0.14** | 11.1 ± 0.23 | 11.42 ± 0.2 | 8.3 ± 0.33 | 6.98 ± 0.23 | 8.48 ± 0.34 |
| ninja | 7.44 ± 0.33 | 7.62 ± 0.2 | **7.72 ± 0.12** | 5.76 ± 0.33 | 5.92 ± 0.2 | 5.7 ± 0.35 |
| plunder | 5.26 ± 0.22 | 5.53 ± 0.12 | **5.57 ± 0.07** | 4.47 ± 0.56 | 4.78 ± 0.36 | 4.56 ± 0.17 |
| starpilot | 20.91 ± 0.51 | 14.39 ± 0.53 | 14.39 ± 0.59 | 25.94 ± 1.68 | **26.36 ± 1.02** | 24.11 ± 0.64 |

Table 5: Mean and Standard Deviation for the **train performances** of each offline learning algorithm averaged over 5 random seeds when trained on **1M Expert Dataset in Procgen**.

| Environment | BC | BCT | DT | BCQ | CQL | IQL |
|---|---|---|---|---|---|---|
| bigfish | 4.38 ± 0.38 | 2.18 ± 0.13 | 2.37 ± 0.16 | 3.57 ± 0.34 | 4.1 ± 0.37 | **4.85 ± 0.52** |
| bossfight | 5.87 ± 0.26 | 0.35 ± 0.11 | 0.52 ± 0.08 | 6.53 ± 0.33 | **8.13 ± 0.13** | 7.62 ± 0.33 |
| caveflyer | **4.92 ± 0.28** | 3.85 ± 0.17 | 3.48 ± 0.35 | 2.15 ± 0.26 | 2.12 ± 0.28 | 3.43 ± 0.22 |
| chaser | **4.62 ± 0.36** | 1.69 ± 0.08 | 1.86 ± 0.04 | 4.05 ± 0.65 | 4.39 ± 0.31 | 3.17 ± 0.17 |
| climber | **4.91 ± 0.22** | 3.49 ± 0.12 | 3.58 ± 0.21 | 0.87 ± 0.12 | 0.98 ± 0.06 | 2.33 ± 0.33 |
| coinrun | **8.26 ± 0.19** | 7.66 ± 0.33 | 8.08 ± 0.22 | 7.02 ± 0.15 | 7.22 ± 0.17 | 7.74 ± 0.21 |
| dodgeball | **0.98 ± 0.07** | 0.89 ± 0.07 | 0.84 ± 0.04 | 0.76 ± 0.09 | 0.85 ± 0.07 | 0.93 ± 0.12 |
| fruitbot | 21.18 ± 0.62 | 13.93 ± 0.75 | 13.56 ± 0.73 | 11.54 ± 2.12 | 16.99 ± 1.55 | **25.22 ± 0.94** |
| heist | **2.42 ± 0.14** | 1.9 ± 0.16 | 1.78 ± 0.09 | 0.48 ± 0.14 | 0.44 ± 0.12 | 0.58 ± 0.26 |
| jumper | **5.68 ± 0.18** | 4.54 ± 0.18 | 5.6 ± 0.23 | 4.14 ± 0.22 | 4.26 ± 0.22 | 4.06 ± 0.21 |
| leaper | **2.84 ± 0.07** | 2.56 ± 0.21 | 2.36 ± 0.18 | 2.48 ± 0.09 | 2.82 ± 0.28 | 2.44 ± 0.21 |
| maze | **4.46 ± 0.16** | 4.26 ± 0.25 | 3.88 ± 0.31 | 2.48 ± 0.11 | 3.04 ± 0.11 | 2.68 ± 0.31 |
| miner | **7.85 ± 0.32** | 1.53 ± 0.06 | 1.47 ± 0.08 | 1.46 ± 0.32 | 2.21 ± 0.24 | 1.66 ± 0.17 |
| ninja | **5.88 ± 0.3** | 5.8 ± 0.07 | 5.74 ± 0.22 | 4.52 ± 0.44 | 4.36 ± 0.25 | 4.38 ± 0.12 |
| plunder | **4.94 ± 0.13** | 4.65 ± 0.12 | 4.7 ± 0.24 | 3.66 ± 0.29 | 3.84 ± 0.23 | 4.03 ± 0.14 |
| starpilot | 17.69 ± 0.59 | 10.9 ± 0.49 | 10.72 ± 0.32 | 22.21 ± 0.7 | 22.42 ± 0.39 | **22.88 ± 0.59** |

Table 6: Mean and Standard Deviation for the **test performances** of each offline learning algorithm averaged over 5 random seeds when trained on **1M Expert Dataset in Procgen**.

> **Generalization to New Environments (Expert)**
>
> In most cases, BC performs similarly or better than the offline RL and sequence modeling approaches on test levels. However, *all the offline learning methods (state-of-the-art offline RL, behavioral cloning, and sequence modeling approaches) perform worse on average than the online RL method (PPO) on unseen environments*. For more than half of the games, offline RL methods cannot reach PPO's test performance. All sequence modelling and offline RL methods fail to generalize on the game of Miner, which is one of the most stochastic games in Procgen.

Tables 5, 6, 7 and 8 show the train and test performances of each offline learning algorithm in 16 Procgen games for both types of offline data regimes, 1M expert and 1M mixed expert-suboptimal respectively. Moreover, we also plot the generalization gap for each algorithm in every game for better understanding of how well they learn from training levels and perform zero-shot on testing levels in Figures 23, 28 and 29.

| Environment | BC | BCT | DT | BCQ | IQL | CQL |
|---|---|---|---|---|---|---|
| bigfish | $7.81 \pm 0.91$ | $7.83 \pm 1.19$ | $7.47 \pm 0.59$ | $7.53 \pm 1.02$ | $\mathbf{8.03 \pm 1.04}$ | $6.53 \pm 0.77$ |
| bossfight | $5.53 \pm 0.22$ | $0.71 \pm 0.1$ | $0.63 \pm 0.07$ | $7.24 \pm 0.58$ | $7.73 \pm 0.1$ | $\mathbf{7.96 \pm 0.44}$ |
| caveflyer | $5.06 \pm 0.44$ | $\mathbf{5.14 \pm 0.06}$ | $4.34 \pm 0.31$ | $3.79 \pm 0.67$ | $2.68 \pm 0.44$ | $3.68 \pm 0.67$ |
| chaser | $\mathbf{4.05 \pm 0.21}$ | $2.37 \pm 0.19$ | $2.35 \pm 0.11$ | $1.74 \pm 0.35$ | $1.42 \pm 0.63$ | $2.76 \pm 0.5$ |
| climber | $\mathbf{6.94 \pm 0.35}$ | $4.9 \pm 0.34$ | $4.99 \pm 0.12$ | $2.0 \pm 0.21$ | $1.15 \pm 0.77$ | $2.98 \pm 0.99$ |
| coinrun | $7.9 \pm 0.47$ | $\mathbf{8.5 \pm 0.06}$ | $8.37 \pm 0.35$ | $7.73 \pm 0.35$ | $6.67 \pm 0.73$ | $7.73 \pm 0.2$ |
| dodgeball | $\mathbf{3.03 \pm 0.05}$ | $3.03 \pm 0.41$ | $2.95 \pm 0.44$ | $2.0 \pm 0.13$ | $2.7 \pm 0.06$ | $2.15 \pm 0.15$ |
| fruitbot | $21.97 \pm 1.12$ | $16.53 \pm 0.93$ | $15.38 \pm 1.09$ | $25.36 \pm 0.29$ | $\mathbf{26.89 \pm 0.28}$ | $26.06 \pm 0.6$ |
| heist | $\mathbf{6.43 \pm 0.34}$ | $4.83 \pm 0.5$ | $4.43 \pm 0.23$ | $2.47 \pm 0.12$ | $1.67 \pm 0.42$ | $1.6 \pm 0.17$ |
| jumper | $\mathbf{7.53 \pm 0.23}$ | $6.43 \pm 0.37$ | $6.93 \pm 0.34$ | $6.83 \pm 0.48$ | $6.93 \pm 0.2$ | $6.87 \pm 0.27$ |
| leaper | $\mathbf{3.1 \pm 0.15}$ | $2.87 \pm 0.28$ | $2.73 \pm 0.27$ | $2.47 \pm 0.09$ | $2.33 \pm 0.68$ | $2.77 \pm 0.03$ |
| maze | $\mathbf{7.8 \pm 0.26}$ | $5.8 \pm 0.2$ | $5.9 \pm 0.65$ | $4.73 \pm 0.17$ | $4.53 \pm 0.48$ | $3.77 \pm 0.03$ |
| miner | $\mathbf{9.49 \pm 0.33}$ | $5.77 \pm 0.69$ | $4.87 \pm 0.34$ | $3.56 \pm 0.24$ | $4.05 \pm 0.67$ | $2.36 \pm 0.41$ |
| ninja | $6.2 \pm 0.1$ | $\mathbf{6.7 \pm 0.17}$ | $6.4 \pm 0.15$ | $5.0 \pm 0.12$ | $3.83 \pm 0.77$ | $4.97 \pm 0.32$ |
| plunder | $\mathbf{5.81 \pm 0.21}$ | $5.41 \pm 0.17$ | $4.9 \pm 0.37$ | $4.18 \pm 0.23$ | $4.34 \pm 0.48$ | $3.93 \pm 0.15$ |
| starpilot | $21.92 \pm 0.35$ | $13.64 \pm 0.12$ | $12.34 \pm 0.57$ | $22.66 \pm 2.14$ | $\mathbf{22.8 \pm 0.84}$ | $21.57 \pm 2.17$ |

Table 7: Mean and Standard Deviation for the train performances of each offline learning algorithm averaged over 3 random seeds when trained on **1M Mixed Expert-Suboptimal Dataset in Procgen**. See Appendix 4.2 for more details.

| Environment | BC | BCT | DT | BCQ | IQL | CQL |
|---|---|---|---|---|---|---|
| bigfish | $2.89 \pm 0.15$ | $2.09 \pm 0.08$ | $2.23 \pm 0.15$ | $4.13 \pm 0.52$ | $\mathbf{4.14 \pm 0.54}$ | $3.64 \pm 0.36$ |
| bossfight | $5.13 \pm 0.14$ | $0.37 \pm 0.18$ | $0.58 \pm 0.16$ | $6.69 \pm 0.57$ | $7.12 \pm 0.43$ | $\mathbf{7.91 \pm 0.35}$ |
| caveflyer | $\mathbf{4.05 \pm 0.24}$ | $3.1 \pm 0.41$ | $3.43 \pm 0.5$ | $2.42 \pm 1.08$ | $1.66 \pm 0.67$ | $1.97 \pm 0.41$ |
| chaser | $\mathbf{3.43 \pm 0.22}$ | $1.95 \pm 0.05$ | $1.86 \pm 0.07$ | $1.44 \pm 0.2$ | $1.41 \pm 0.6$ | $2.58 \pm 0.12$ |
| climber | $\mathbf{4.64 \pm 0.29}$ | $3.18 \pm 0.21$ | $2.93 \pm 0.42$ | $0.73 \pm 0.25$ | $0.57 \pm 0.35$ | $0.94 \pm 0.14$ |
| coinrun | $\mathbf{7.77 \pm 0.24}$ | $7.47 \pm 0.07$ | $7.5 \pm 0.2$ | $6.63 \pm 0.12$ | $6.0 \pm 0.36$ | $7.17 \pm 0.41$ |
| dodgeball | $\mathbf{1.19 \pm 0.14}$ | $1.13 \pm 0.05$ | $0.94 \pm 0.09$ | $0.55 \pm 0.16$ | $0.87 \pm 0.11$ | $0.67 \pm 0.01$ |
| fruitbot | $18.84 \pm 0.7$ | $11.39 \pm 0.61$ | $11.35 \pm 1.39$ | $22.76 \pm 1.44$ | $22.0 \pm 0.43$ | $\mathbf{24.46 \pm 0.52}$ |
| heist | $\mathbf{2.37 \pm 0.3}$ | $2.0 \pm 0.17$ | $1.97 \pm 0.07$ | $0.53 \pm 0.15$ | $0.27 \pm 0.03$ | $0.5 \pm 0.06$ |
| jumper | $\mathbf{4.63 \pm 0.47}$ | $4.27 \pm 0.43$ | $4.53 \pm 0.3$ | $2.8 \pm 0.15$ | $3.0 \pm 0.5$ | $2.9 \pm 0.17$ |
| leaper | $2.6 \pm 0.25$ | $2.5 \pm 0.06$ | $\mathbf{2.7 \pm 0.35}$ | $2.43 \pm 0.29$ | $2.27 \pm 0.53$ | $2.43 \pm 0.26$ |
| maze | $4.77 \pm 0.32$ | $4.77 \pm 0.19$ | $\mathbf{5.5 \pm 0.06}$ | $1.93 \pm 0.13$ | $2.1 \pm 0.15$ | $1.97 \pm 0.18$ |
| miner | $\mathbf{6.56 \pm 0.09}$ | $1.28 \pm 0.11$ | $1.28 \pm 0.08$ | $0.51 \pm 0.15$ | $0.8 \pm 0.1$ | $0.43 \pm 0.08$ |
| ninja | $\mathbf{5.23 \pm 0.12}$ | $5.37 \pm 0.38$ | $5.1 \pm 0.36$ | $4.67 \pm 0.43$ | $3.23 \pm 0.81$ | $3.77 \pm 0.26$ |
| plunder | $\mathbf{4.59 \pm 0.16}$ | $4.53 \pm 0.16$ | $4.3 \pm 0.17$ | $3.39 \pm 0.28$ | $3.86 \pm 0.25$ | $3.76 \pm 0.26$ |
| starpilot | $17.93 \pm 0.32$ | $11.64 \pm 0.71$ | $10.69 \pm 0.23$ | $\mathbf{21.86 \pm 2.07}$ | $19.64 \pm 1.79$ | $20.11 \pm 0.43$ |

Table 8: Mean and Standard Deviation for the test performances of each offline learning algorithm averaged over 3 random seeds when trained on **1M Mixed Expert-Suboptimal Dataset in Procgen**. See Appendix 4.2 for more details.

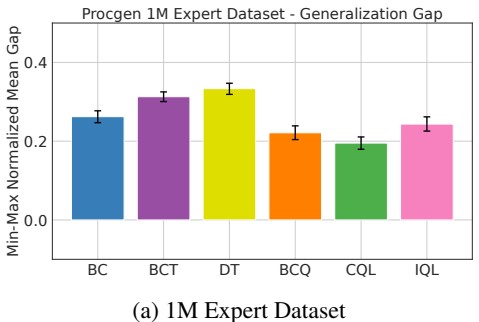

(a) 1M Expert Dataset

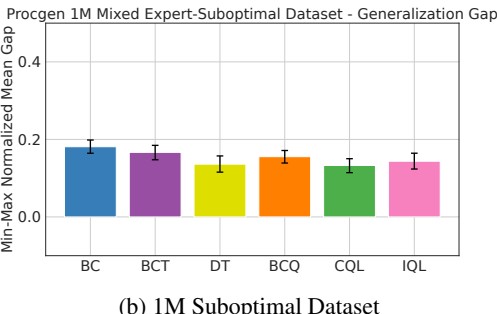

(b) 1M Suboptimal Dataset

Figure 23: Min-max normalized mean **Generalization Gap** plots for offline learning algorithms trained using 1M expert and suboptimal dataset in Procgen respectively.

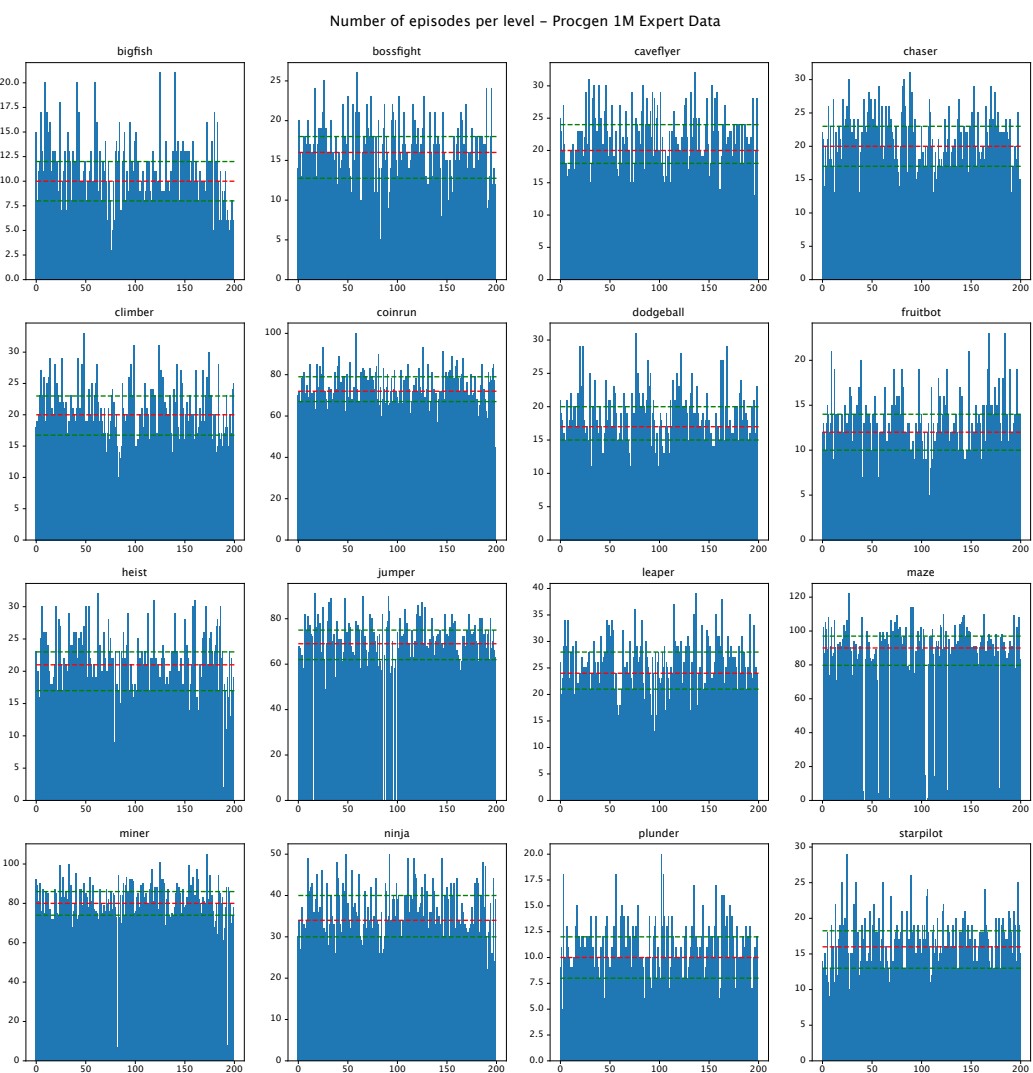

Figure 24: Number of episodes per level from the training set for each environment in the 1M expert dataset from Procgen. Red line represents the median. Lower and upper green lines represent the 25th and 75th percentile of the values respectively.

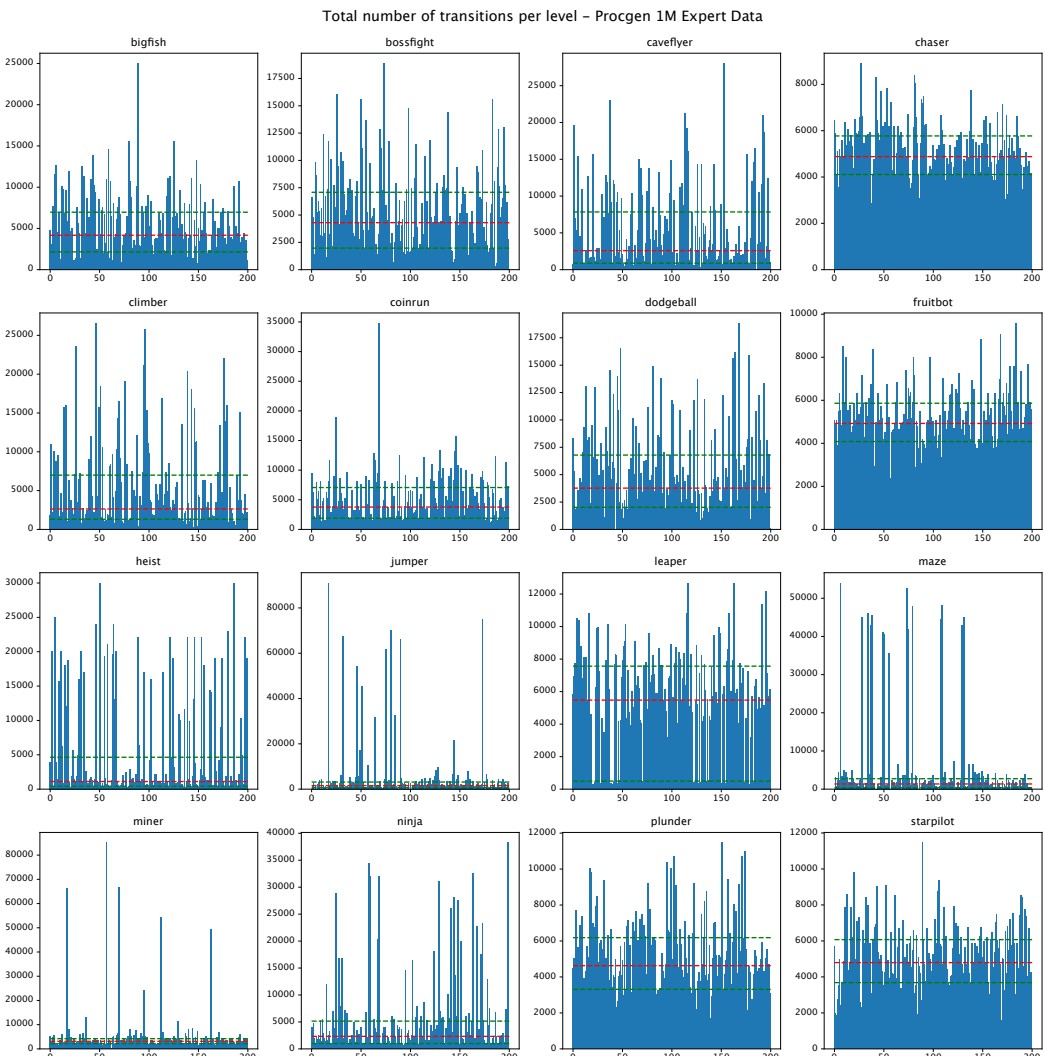

Figure 25: Total number of transitions per level from the training set for each environment in the 1M expert dataset from Procgen. Red line represents the median. Lower and upper green lines represent the 25th and 75th percentile of the values respectively.

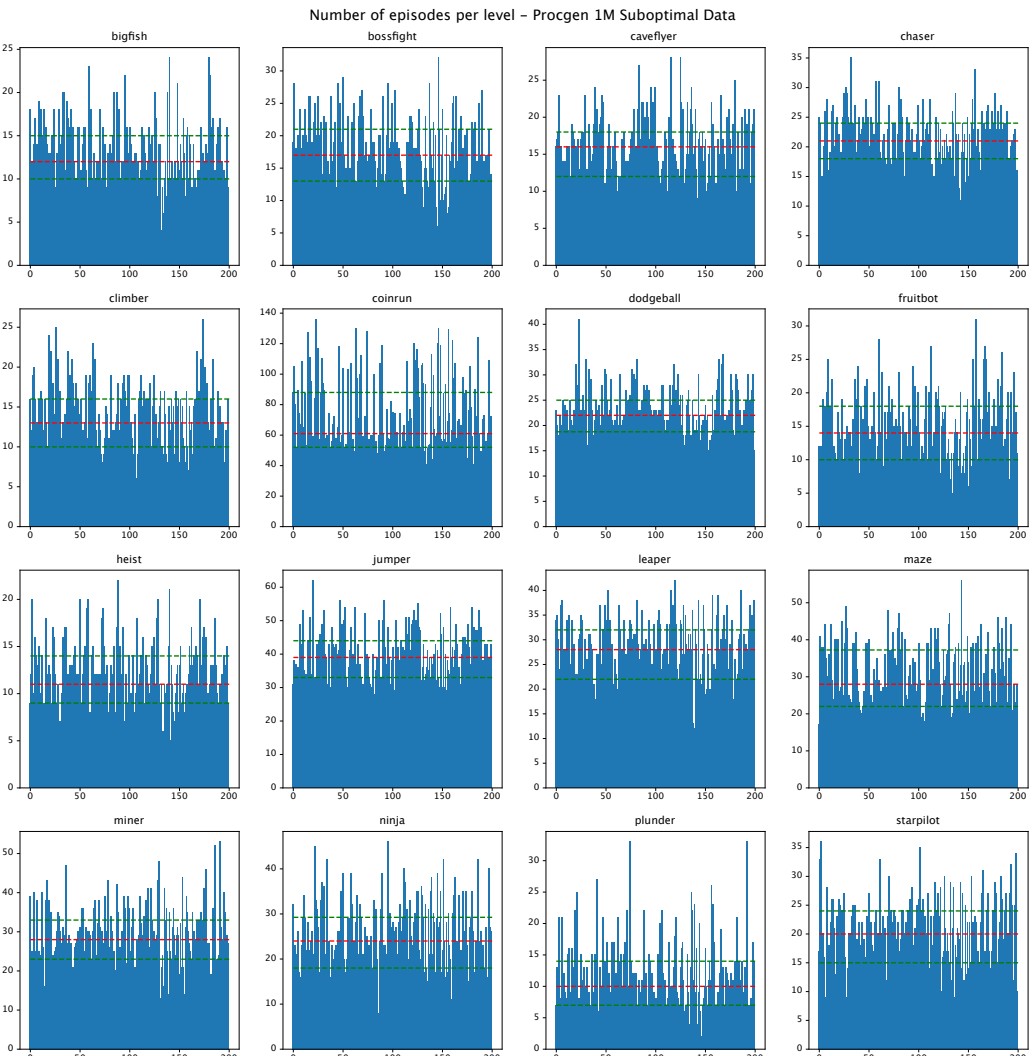

Figure 26: Number of episodes per level from the training set for each environment in the 1M suboptimal dataset from Procgen. Red line represents the median. Lower and upper green lines represent the 25th and 75th percentile of the values respectively.

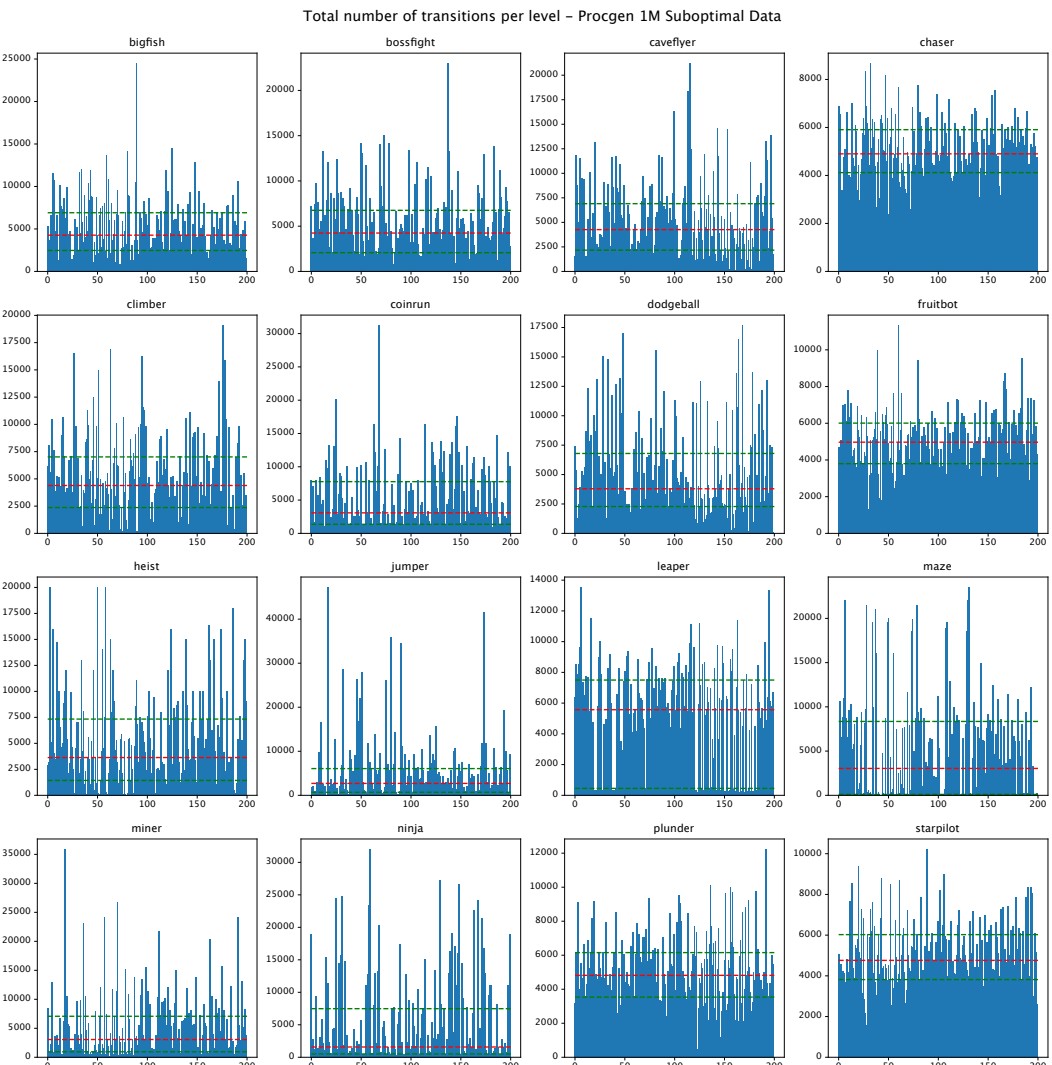

Figure 27: Total number of transitions per level from the training set for each environment in the 1M suboptimal dataset from Procgen. Red line represents the median. Lower and upper green lines represent the 25th and 75th percentile of the values respectively.

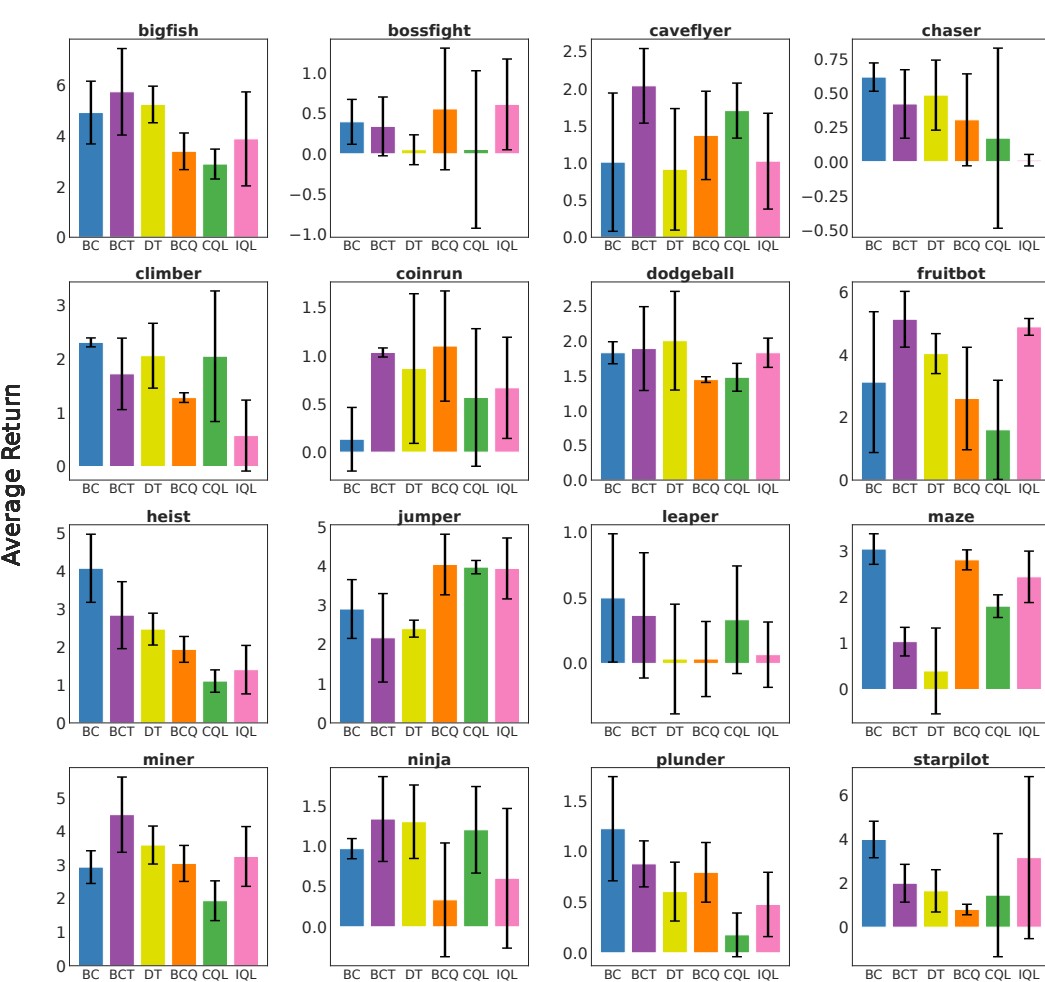

Figure 28: Mean and standard deviation of Per-game Generalization Gap for every offline learning algorithm in Procgen when trained using 1M suboptimal data. Every algorithm is run for 3 random trials.

Figure 29: Mean and standard deviation of Per-game Generalization Gap for every offline learning algorithm in Procgen when trained using 1M expert data. Every algorithm is run for 5 random trials.

