# OpenReview forum: "The Generalization Gap in Offline Reinforcement Learning"
_ICLR.cc/2024/Conference — ICLR 2024 poster_

### Official Review · Reviewer_qttr · 2023-10-24

**Soundness:** 3 good
**Presentation:** 3 good
**Contribution:** 3 good
**Rating:** 6
**Confidence:** 4

**Summary:**

This paper studies the problem of generalization for popular offline RL algorithms. By carrying out extensive experiments of state-of-the-art offline RL algorithms (IQL, CQL, DT, etc) on procgen and webshop environments, this paper presents the findings that although offline RL methods outperform BC in training environments with suboptimal data, they fail to generalize as well to testing environments similar to training environments.

**Strengths:**

This paper carries out extensive experiments on procgen and another more realistic environment webshop, covering most state-of-the-art offline rl methods such as CQL, IQL and BCQ.

The problem of generalization of offline reinforcement learning is important yet relatively scarcely touched.

The paper is well-written and easy-to-follow. The format of using red text boxes makes it easy to capture important takeaways.

**Weaknesses:**

I have concerns that some of the offline RL baselines might not be tuned appropriately. For example, in the procgen environments, there seems to be a large gap between BC and CQL both in the train and test environment. However, CQL contains a weighted combination of TD-learning loss and behavior cloning loss so a proper tuning of the weights should make CQL at least comparable to BC.

Although procgen is a relatively popular benchmark, it is not the primary benchmark for those offline RL methods. They are most extensively tested on continuous control tasks such as D4RL (https://arxiv.org/abs/2004.07219), so authors should explain why they did not conduct experiments in the setting of continuous control. For example, the different test environments can be attained by modifying the environment parameters such as gravity.

The conclusion of the paper is derived mostly through empirical observations. It would be important to also have theoretical understandings of why offline RL does not work as well as BC in terms of generalization, similar to the analysis done in https://arxiv.org/pdf/2204.05618.pdf.

I am willing to raise my score if those concerns can be solved.

**Questions:**

Why IQL cannot be used in the setting of webshop? The expectile regression part should not depend on the number of actions for each state and as long as the expectile regression can be implemented it should be fine?

In section 4.5, is it possible to extend the plot a bit to show around how much data the learning curve stops to grow? It seems that we also need more than 3 data points to draw a valid conclusion regards to the trend.

In Figure 3, why is the blue line sometimes higher than the red line? Isn't the data collected by expert PPO?

---

> ### Author Response · Authors · 2023-11-17
> **Response from authors**
>
> We are grateful for your valuable feedback and hope our answers below will address your remaining concerns. We are pleased to see that you found our paper to be “well-written, easy-to-follow” and the topic important.
>
> > I have concerns that some of the offline RL baselines might not be tuned appropriately.
>
> We have included thorough details on all the parameters that we had swept for tuning all our baselines in **Appendix G.2**
>
> To summarize, for Procgen:
>
>         “For BC, we performed a sweep over the batch size ∈ {64,128,256,512} transitions and learning rate ∈ {5e − 3, 1e − 4, 5e − 4, 6e − 5}. For BCQ, CQL, and IQL, which use a DQN-style training setup (i.e., they have a base model and a frozen target model) (Mnih et al., 2013), in addition to the hyperparameters mentioned for BC, we swept over whether to use polyak moving average or directly copy weights, in the latter case, the target model update frequency ∈ {1, 100, 1000} and in the former case, the polyak moving average constant τ ∈ {0.005, 0.5, 0.99}. For BCQ, we also swept over the threshold value for action selection ∈ {0.3, 0.5, 0.7}. For CQL, we swept over the CQL loss coefficient, which we refer to as cql alpha in our codebase, ∈ {0.5, 1.0, 4.0, 8.0}. Finally, for IQL, we sweep over the temperature ∈ {3.0, 7.0, 10.0} and the expectile weight ∈ {0.7, 0.8, 0.9}.
>
>         For sequence modelling algorithms, DT and BCT, we sweep over the learning rate and batch size mentioned above, as well as the context length size ∈ {5, 10, 30, 50}. For DT, we also sweep over the return-to-go (rtg) multiplier ∈ {1, 5}. We follow similar approach in (Chen et al., 2021) to set the maximum return-to-go at inference time by finding the maximum return in the training dataset for a particular game and then multiplying by either 1 or 5 depending on the rtg multiplier value. We also use the default value of 0.1 for dropout in DT and BCT from (Chen et al., 2021).”
>
> **For CQL, we swept over 4 (batch size) * 5 (learning rate) * 3 (update freq.) * 3 (tau)  * 4 (cql alpha)=720 different combinations**, with each combination trained and evaluated thrice for different random seed, amounting to a total of **2160 training runs for tuning CQL**. Similar is the case for other baselines.
>
> **We have also run a more comprehensive sweep for 10M expert dataset and have updated Table 2 in the Appendix accordingly. In WebShop too we ran a similar comprehensive sweep which is mentioned in Appendix G.2.2.**
>
> In conclusion, **we believe we’ve done extensive HP tuning with similar budgets for all our baselines.**
>
> > D4RL (https://arxiv.org/abs/2004.07219), so authors should explain why they did not conduct experiments in the setting of continuous control. For example, the different test environments can be attained by modifying the environment parameters such as gravity.
>
> Thank you for such an insightful question. First of all, **we would like to emphasize that D4RL is not a standard benchmark for testing generalization. Procgen was chosen because it is one of the most commonly used benchmarks for this purpose in the online RL setting**, and therefore we wanted to advocate its use for offline setting as well. Similarly, WebShop is an equally challenging generalization benchmark in a different domain which is relevant for real-world applications (particularly of language models interacting with the web, which is a timely subject of wide interest to the community).
>
> Moreover, the offline RL methods that we evaluated are advertised for both, continuous as well as discrete domains. BCQ, CQL and DT especially have achieved state-of-the-art results on Atari at the time of their papers’ acceptance.
>
> **Our research, on the other hand, emphasizes the importance of testing offline RL methods across a diverse range of tasks. While these methods are often presented as universally applicable, identifying scenarios where they falter is crucial. This approach not only highlights the limitations of current methods but also encourages the development of more robust and versatile solutions.**

---

> > ### Author Response · Authors · 2023-11-17
> > **Response from authors (contd...)**
> >
> > > conclusion of the paper is derived mostly through empirical observations. It would be important to also have theoretical understandings of why offline RL does not work as well as BC in terms of generalization
> >
> > While we acknowledge the importance of theoretical findings, **we believe that the absence of theoretical results in this context should not be a basis for rejection.** Notably, several highly-regarded papers, including those that have received best paper awards or orals, have been empirical in nature [1]. In addition, **theory for generalization to new tasks of offline RL algorithms is an understudied and non-trivial subject so we believe it would be better addressed in a separate paper.**
> >
> > However, **we hypothesize the following factors could be why online RL > BC > offline RL:**
> > - Prior research [2] indicates that offline RL can outperform BC under very specific conditions, such as sparse rewards or noisy data. In our study, however, these conditions do not prevalently exist as our data is relatively noise-free and the reward is not excessively sparse.
> > - It is known that offline RL methods tend to be overly cautious with actions not previously encountered in the training dataset. When evaluated in new environments, this approach often results in suboptimal actions, as the methods are excessively risk-averse. Contrastingly, BC benefits from a broader action space, allowing for better generalization through learned representations, without the constraints imposed by offline RL methods.
> > - In some cases, offline RL might devalue all actions at test time if they are considered out-of-distribution, leading to a quasi-random and less effective policy. This phenomenon can significantly hinder the method's effectiveness in new environments. [3]
> > - We also refer to [4] to support our argument that online RL generally outperforms offline learning methods in terms of generalization. This is due to its ability to collect its own diverse data and find (optimal) actions that maximize reward, rather than imitating potentially suboptimal actions, as is the case with BC trained on data from various sources.
> >
> >
> > [1]: Kumar, A., Agarwal, R., Geng, X., Tucker, G., & Levine, S. (2023). Offline q-learning on diverse multi-task data both scales and generalizes. International Conference on Learning Representations (ICLR) 2023.
> >
> > [2]: Kumar, A., Hong, J., Singh, A., & Levine, S. (2022). When should we prefer offline reinforcement learning over behavioral cloning?. *arXiv preprint arXiv:2204.05618*.
> >
> > [3]: Ghosh, D., Ajay, A., Agrawal, P., & Levine, S. (2022, June). Offline rl policies should be trained to be adaptive. In *International Conference on Machine Learning* (pp. 7513-7530). PMLR.
> >
> > [4]: Xu, T., Li, Z., Yu, Y., & Luo, Z. Q. (2021). On generalization of adversarial imitation learning and beyond. arXiv preprint arXiv:2106.10424.
> >
> > > Why IQL cannot be used in the setting of webshop?
> >
> > **Due to the time constraints and the limited computational resources available during the rebuttal phase, we are unable to include IQL in our analysis while ensuring robust tuning of hyperparameters.** The complexity of the method and its sensitivity to hyperparameters necessitate substantial effort for accurate integration in our codebase. **We aim to incorporate IQL in our code by the camera-ready submission phase if the paper is accepted.**
> >
> > However, it is important to note that IQL is the least effective method on Procgen among the ones we evaluate. Additionally, the other offline RL methods have not shown promising results in the WebShop domain. Hence, **we anticipate adding IQL is unlikely to alter our main conclusions, as the performance ranking of  offline RL methods appear consistent across Procgen and WebShop.**
> >
> > > In section 4.5, is it possible to extend the plot a bit to show around how much data the learning curve stops to grow?
> >
> > **We cannot do this due to a limitation with our computational resources.** Specifically, the V100 GPUs we utilized are not capable of supporting experiments that have > 10M transitions. This constraint is the primary reason we cannot extend the learning curve further in our analysis.
> >
> > However, **we have open-sourced our 25M suboptimal dataset and encourage contributions from the broader research community, which may have access to more powerful computational resources.** By doing so, we hope to facilitate further investigation into how the generalization capabilities scale.

---

> > > ### Author Response · Authors · 2023-11-17
> > > **Response from authors (contd...)**
> > >
> > > > In Figure 3, why is the blue line sometimes higher than the red line? Isn't the data collected by expert PPO?
> > >
> > > The blue line (representing the dataset average) is sometimes higher than the red line (representing PPO performance) in Figure 3 **due to the difference in sample sizes used for calculations.** The PPO line is derived from evaluating the policy for only 100 episodes. In contrast, the dataset average, which is a more robust comparison point, encompasses data from over >1000 episodes. This **larger sample size not only provides a more accurate estimate but is also better point of comparison because offline RL baselines learn from this dataset**. Therefore, it's quite common for the dataset average to occasionally outperform PPO.
> > >
> > > We hope that our responses have adequately addressed your concerns. If you could consider increasing the score for our submission, we would greatly appreciate it. Please let us know if any further concerns are impacting your decision.

---

> > > > ### Author Response · Authors · 2023-11-20
> > > > **Possible reminder and re-evaluation of scores**
> > > >
> > > > Dear Reviewer,
> > > >
> > > > Your thorough review and insightful comments have been instrumental in enhancing the quality of our paper.
> > > >
> > > > Since the rebuttal deadline is near, if any remaining points need further attention, we are eager to make those improvements. If you find our responses satisfactory, we would be grateful for a reconsideration of your scoring.

---

> > > > > ### Comment · Reviewer_qttr · 2023-11-20
> > > > > **Thanks for the rebuttal**
> > > > >
> > > > > Thanks authors for the rebuttal. I do understand the common use of procgen to study the generalization of reinforcement learning algorithms and emphasize with the authors about the limited computational resources for more experiments. However, I still think the proposed experiments and theoretical understandings are important to further support the conclusion of the paper. Therefore, I decide to keep my score of weak accept.

---

### Official Review · Reviewer_4tcZ · 2023-10-27

**Soundness:** 3 good
**Presentation:** 3 good
**Contribution:** 3 good
**Rating:** 6
**Confidence:** 4

**Summary:**

This paper study and benchmark the generalization abilities of existing offline RL algorithms, revealing several interesting and helpful conclusions regarding to current offline RL learning researches.

**Strengths:**

1. The problem studied in this paper is important and interesting. Existing offline RL benchmarks indeed require more practical metrics for evaluation.
2. The experiments conducted in this paper seem solid and abundant. The conclusions made sound convincing.
3. Hyperparameters are provided and the results seem reproducible.

**Weaknesses:**

1. The algorithms included are a bit limited. Methods like model-based learning [1-2], curriculum imitation [3], and other methods are not involved. More useful conclusions can be made when the benchmarking algorithms are expanded.

2. Benchmark included is a bit limited. Most of the results are concluded from a simulated benchmark ProcGen, only a small part of experiments are conducted on the real-world dataset "WebShop". The author may consider expanding their tested benchmark to more real-world problems as introduced in [4].

[1] Yu T, Thomas G, Yu L, et al. Mopo: Model-based offline policy optimization[J]. Advances in Neural Information Processing Systems, 2020, 33: 14129-14142.
[2] Yu T, Kumar A, Rafailov R, et al. Combo: Conservative offline model-based policy optimization[J]. Advances in neural information processing systems, 2021, 34: 28954-28967.
[3] Liu M, Zhao H, Yang Z, et al. Curriculum offline imitating learning[J]. Advances in Neural Information Processing Systems, 2021, 34: 6266-6277.
[4] Qin R J, Zhang X, Gao S, et al. NeoRL: A near real-world benchmark for offline reinforcement learning[J]. Advances in Neural Information Processing Systems, 2022, 35: 24753-24765.

**Questions:**

1. Will the author open-source their benchmark and evaluation codes?
2. Can the author explain more about the setup of Webshop? What is the state/action/reward/objective/transition of this problem? An example would be better (can lie in the appendix).
3. Can you explain how you fine-tune these algorithms and how the results are selected? It is thorny and important to select the best model in practice.

---

> ### Author Response · Authors · 2023-11-17
> **Response from authors (1/2)**
>
> We would like to thank the reviewer for providing us with valuable feedback. We are glad that the reviewer found our paper’s results to be “important and interesting” and our experiments to be “solid and abundant”.
>
> > The algorithms included are a bit limited. Methods like model-based learning [1-2], curriculum imitation [3], and other methods are not involved. More useful conclusions can be made when the benchmarking algorithms are expanded.
>
> Our paper aims to evaluate the most commonly used offline RL algorithms on a new benchmark that probes their generalization abilities (which is an understudied problem in the literature). We believe we have accomplished this by benchmarking BCQ, CQL, IQL, DT, BCT, BC etc., which have a combined 3000+ citations and are competitive on a wide range of commonly used offline RL benchmarks [1]. In contrast, the methods you referenced are rather specialized and don’t seem to be standard baselines in common offline RL papers. There are a lot of offline RL algorithms and we cannot possibly implement, tune, and evaluate all of them on a new benchmark (especially if we want to do justice to all these methods via extensive HP tuning using similarly large budgets to ensure fair comparisons—which we do for our current baselines in Appendix G). **Since we had to draw the line somewhere, we decided to include only the most versatile, effective, and widely used offline learning methods according to the literature.**
>
> **Our goal was to create a benchmark that is accessible and easy to use by the community with simple and unified implementations** in order to speed up progress. Therefore, **we have open-sourced our code and datasets, and we welcome contributions from everyone to integrate and tune their baselines on our benchmark.**
>
> [1]: Tarasov, D., Nikulin, A., Akimov, D., Kurenkov, V., & Kolesnikov, S. (2022). CORL: Research-oriented deep offline reinforcement learning library. arXiv preprint arXiv:2210.07105.
>
> > Most of the results are concluded from a simulated benchmark ProcGen, only a small part of experiments are conducted on the real-world dataset "WebShop". The author may consider expanding their tested benchmark to more real-world problems as introduced in [4].
>
> While we covered a significant portion of our experiments on Procgen in the main paper, **we covered similar experiments conducted with the WebShop dataset in Appendix N of our submission**. Here we present a comprehensive assessment on various sizes of the WebShop dataset (i.e. 100, 500, 1000, 1500, 5000, 10000 episodes), as well as study the impact of suboptimal (collected from IL policy) versus expert data (collected from humans) in WebShop. These experiments are also crucial in demonstrating the robustness and adaptability of our approach in a real-world context.
>
> Furthermore, we want to emphasize that **our research is highly relevant and timely, particularly with the increasing importance of LLM-based agents interacting with the internet**. The Procgen benchmark, which learns directly from high-level images in an applied domain (games), and the WebShop environment, which is a realistic simulator of e-commerce websites, both are highly relevant and practical. Specifically, our experiments with WebShop showcase generalization to real-world scenarios, including interactions with actual websites like Amazon.
>
> **While NeoRL [4] does not assess generalization to new levels/instructions, the WebShop simulator on the other hand allows a realistic emulation of web interactions, thus enabling the agents to generalize to real-world scenarios like the Amazon website**. We argue that this is a substantial step towards real-world applicability.
>
> We do not currently plan to expand the benchmark to other domains for the purpose of this submission. However, we are open to exploring this in the future and will aim to make it easy for others to contribute additional datasets since **we will be open-sourcing our codebase. Both our repositories can be adapted to other similar environments i.e. image-based/state-based environments in the Procgen repository, and text-based environments in the WebShop repository.**
>
> > Will the author open-source their benchmark and evaluation codes?
>
> We would like to clarify that both, **the code and the datasets, are already uploaded as zip files as part of the supplementary material here and we will definitely open-source the code upon acceptance**.
>
> **We answer the remaining questions in the next comment.**

---

> > ### Author Response · Authors · 2023-11-17
> > **Response from authors (2/2)**
> >
> > > Can the author explain more about the setup of Webshop? What is the state/action/reward/objective/transition of this problem? An example would be better (can lie in the appendix).
> >
> > We appreciate the reviewer for bringing this up. We have now **expanded Appendix F.2 with additional details on the WebShop environment.** We also add those details here for your reference:
> >
> >         A single transition in WebShop comprises of the following:
> >
> >         (1) State: A state in WebShop is defined by the type of webpage the agent is currently viewing. This can be one of four types: a search page with a search bar, a results page displaying search outcomes, an item page detailing a product, or an item-detail page offering more extensive information about a product.
> >         (2) Observation: The agent is actually fed with a more minimal, processed version of underlying HTML source code of the page along with the jpeg images that are present on that webpage.
> >         (3) Action: Two types of actions are available to the agent: (1) on search page, searching with a specific text query (e.g., search[White shirt]) (2) on all other webpages, choosing an option presented as a text button (e.g., choose[Size M]).
> >         (4) Reward: Once the agent selects the buy action, rewards are computed based on how well the selected product's attributes, type, options, and price match relative to the given instruction. For more details on how the reward is calculated, refer to Section 3.1 in WebShop's paper.
> >         (5) Done: A boolean flag indicating end of an episode, which returns True either when the agent selects "choose[Buy]" action or when the agent exceeds 100 timesteps.
> >
> > > Can you explain how you fine-tune these algorithms and how the results are selected? It is thorny and important to select the best model in practice.
> >
> > We have mentioned the details extensively in **Appendix G.2**. To summarize, for each dataset type, we ran **many different combinations of hyperparameters (for e.g., in CQL, we ran 720 combinations!), which are mentioned in Tables 2 and 3** in the Appendix, for 3 random seeds and then selected the best-performing model by computing the min-max normalized mean train and validation performance for each hyperparameter combination. We also used early stopping to make sure that our selected hyperparameters were not overfitting to the training data.
> >
> > We hope that our responses have adequately addressed your concerns. It would be greatly appreciated if you could consider increasing the score for our submission and let us know if there are any further concerns that might be impacting your decision

---

> > > ### Author Response · Authors · 2023-11-20
> > > **Possible reminder and re-evaluation of score**
> > >
> > > Dear Reviewer,
> > >
> > > We are truly grateful for the time and effort you invested in reviewing our paper. Your comments have been valuable in refining our submission.
> > >
> > > Should any outstanding concerns or questions remain, we are more than willing to address them. If you are satisfied with the revisions made and our responses to your questions, we would be grateful if you could consider revising your score to reflect these improvements.

---

> > ### Comment · Reviewer_4tcZ · 2023-11-22
> > **Thanks for your reply**
> >
> > 1. "the methods you referenced are rather specialized and don’t seem to be standard baselines in common offline RL papers", I disagree, why model-based and imitation-based methods is specialized? "draw the line somewhere", what line? It is too subjective.
> >
> > 2. Thanks for the explanation of the WebShop dataset.
> >
> > 3. About model selection. It seems that you are using the test time effect to select the best checkpoint, this may not be feasible in real-world problems, I think there should be a practical way to make a fair comparison (by fair I mean not only the best results are reported)
> >
> > 4. Thanks for your commitment to open-source, I really appreciate that.
> >
> > 5. Do not push me to increasing the score, which should be a natural thing after my concerns are eased, and is not your job.

---

> > > ### Author Response · Authors · 2023-11-22
> > > **Response to reviewer**
> > >
> > > Thank you for your response. Please find below our reply to your concerns:
> > >
> > >  > "the methods you referenced are rather specialized and don’t seem to be standard baselines in common offline RL papers", I disagree, why model-based and imitation-based methods is specialized? "draw the line somewhere", what line? It is too subjective.
> > >
> > > Regarding the use of model-based and imitation-based methods, we did not mean to imply these methods are overly specialized. We simply wanted to explain our rationale for how we selected what methods to evaluate. We decided to evaluate a few of the most popular methods from different classes of algorithms: imitation learning, sequence modeling, and offline RL. We wanted to prioritize the most commonly used methods in the literature in order to enhance our work’s impact. Unfortunately, we are limited by time and computational resources so we cannot implement all possible approaches, so we had to draw the line somewhere.
> > >
> > > Nonetheless, given that our code is open source, we leave the incorporation and analysis of these additional methods for future research and also encourage researchers working on the above-mentioned domains to benchmark their models on our datasets.
> > >
> > > > About model selection. It seems that you are using the test time effect to select the best checkpoint, this may not be feasible in real-world problems, I think there should be a practical way to make a fair comparison (by fair I mean not only the best results are reported)
> > >
> > > Our methodology  employs an online validation set to select the best checkpoint which is distinct from the test set used to report the results. We believe this approach is both practical and fair, as it aligns with common practices in the field. This validation set can simply be a small percentage (~5-20%) of the training levels, without assuming any access to the test set.
> > >
> > > Can you provide more details on why you think this is not feasible in real-world problems? We believe it is and in fact, we recommend the use of a validation set for model selection (by splitting the entire dataset or simulation environments into train and validation and performing model selection either online or offline depending on the setting). We are open to suggestions on alternative methods that you might consider more appropriate for a fair comparison, and would greatly value any specific recommendations you might have.
> > > If you would like us to report results across the entire set of hyperparameters in order to do a sensitivity analysis, we are happy to include that for the camera ready.
> > >
> > >
> > > > Do not push me to increasing the score, which should be a natural thing after my concerns are eased, and is not your job.
> > >
> > > We apologize if our previous communication seemed to be pushing for an increased score. Our intention was to ensure that we have fully understood and addressed your concerns, and to inquire if there are any remaining issues  so that we have a chance of addressing them before the end of the rebuttal. We appreciate your guidance and feedback in this process, which have already improved our paper.

---

### Official Review · Reviewer_HieC · 2023-10-28

**Soundness:** 2 fair
**Presentation:** 2 fair
**Contribution:** 2 fair
**Rating:** 6
**Confidence:** 4

**Summary:**

This paper studies the generalization abilities of offline RL algorithms across different environments. In particular, it introduced a collection of offline RL datasets of different sizes and skill-levels from the Procgen and WebShop environments. Experiments show that existing offline RL methods perform significantly worse than online RL on both train and test environments. Additional experiments show that an increase in data diversity improves generalization while an increase of the size of training data does not.

**Strengths:**

- This paper investigates an important problem in offline RL.

- It introduced a collection of offline RL datasets of different sizes and skill-levels from the Procgen and WebShop environments.

- The experiments are thorough.

- The writing is clear and easy to follow.

**Weaknesses:**

- The novelty of the study is somewhat restricted. Given that many existing offline RL algorithms do not inherently prioritize generalization ability in their design, so the current experimental results are mostly within expected outcomes.

- There are many duplicate references: "Leveraging procedural generation to benchmark reinforcement learning", "Offline q- learning on diverse multi-task data both scales and generalizes", "Deep residual learning for image recognition.", "The nethack learning environment."

- This paper primarily serves as a summary of empirical observations, and no specific solutions are proposed to address the generalization issue. IMHO, there is a lack of deeper understanding of the generalization issue of offline RL agents and the take aways for readers from this work is limited.

**Questions:**

I appreciate the authors' efforts to investigate an important question in offline RL. However, upon reviewing the current draft, I find the perspective presented to be somewhat one-sided. It is essential to acknowledge that the overall conclusions drawn in the paper might be confined to the specific benchmark dataset being utilized. There have been notable instances demonstrating the impressive generalization capabilities of offline RL agents [1] [2]. Consequently, it would be prudent to avoid overly definitive statements in this paper, given these successful examples.

Moreover, I would like to highlight a gap in the current work—there is a lack of more in-depth analysis to elucidate the key question: "why certain offline RL agents [1][2] exhibit superior generalization skills while others do not?"

A more profound theoretical analysis might be instrumental in explaining these discrepancies.

[1] (Agarwal et al., 2020) An Optimistic Perspective on Offline Reinforcement Learning

[2] (Kumar et al., 2023) Offline Q-Learning on Diverse Multi-task Data Both Scales and Generalizes

---

> ### Author Response · Authors · 2023-11-17
> **Response from authors**
>
> Thank you for providing valuable and constructive feedback which greatly improves our work. We were glad you found our “experiments thorough” and “writing clear and easy-to-follow”.  Below we address the points raised in the same order:
>
> > novelty is restricted. Given that many existing offline RL algorithms do not inherently prioritize generalization ability in their design, so the current experimental results are mostly within expected outcomes.
>
> We respectfully disagree with this assessment of our contributions. As far as we know, **our paper is the first to introduce a benchmark specifically focused on evaluating the generalization of offline RL algorithms.** The absence of such a benchmark limits our understanding of these algorithms' real-world applicability, and our work strives to bridge this gap. In addition, we believe our results are novel in that this is the **first paper to demonstrate that online RL > behavioral cloning > sequence modeling > offline RL approaches when it comes to generalization to new tasks**. Even if our results weren’t surprising, **we believe it is important to show this empirically via a thorough comparison study that others can later reproduce and build on since, as we show in the paper, details such as the data diversity or size can lead to different results**. Moreover, our findings emphasize the need for more robust algorithms suitable for practical applications and establish a standardized benchmark for the equitable assessment of these methods across diverse scales and qualities. Hence, **our contributions consist of: the creation of multiple datasets from two different domains, along with the extensive evaluation of multiple algorithms, and a clear evaluation protocol**.
>
> > many duplicate references
>
> Thank you for pointing this out! We have now rectified the references.
>
> > paper primarily serves as a summary of empirical observations, and no specific solutions are proposed to address the generalization issue
>
> While we agree that specific techniques (either new or existing ones such as regularization) could increase performance of these offline RL, the **purpose of our work isn’t to improve existing offline RL algorithms but rather to thoroughly evaluate *existing* ones as they are typically used in the literature (and in practice)**.
>
> > Consequently, it would be prudent to avoid overly definitive statements in this paper, given these successful examples…… Moreover, I would like to highlight a gap in the current work—there is a lack of more in-depth analysis to elucidate the key question: "why certain offline RL agents [1][2] exhibit superior generalization skills while others do not?"
>
> We acknowledge the points you raised and we will revise our manuscript to address them. We agree that our conclusions should be more cautiously phrased, emphasizing that they are based on evaluations within specific domains. Indeed, there's a need for further research to study the reason behind the good/bad generalizability of these methods.
>
> Furthermore, we would like to re-emphesize the following:
> 1. Paper [1] doesn't have any experiments on generalization to new environments so it is **not a relevant comparison to our work.**
> 2. This is also mentioned in the **“Extended Related Works” section in Appendix D of our submission**, i.e. methods like [2] study how to generalize skills across different Atari games, using a dataset of about 40 games (and [1] only studies online generalization to the same game). In contrast, our research is about how well different levels or instructions within the same game can be handled, and we do this within the Procgen as well as the WebShop framework. Moreover, **our dataset and benchmark is designed to be memory-efficient, making it easier for the academic community to use without needing many GPUs. This is unlike the approaches in [2], which require a large number of TPUs and are practically infeasible to reproduce in most of the academic settings**.
>
> We hope our responses above have adequately addressed your queries and that you will consider increasing your score. Please let us know if there is anything else preventing you from recommending acceptance of our paper.

---

> > ### Author Response · Authors · 2023-11-20
> > **Possible reminder and re-evaluation of score**
> >
> > Dear Reviewer,
> >
> > We greatly appreciate the time and effort you've dedicated to reviewing our paper. Your insightful feedback has been invaluable in refining our work. We hope you find our responses and revisions satisfactory.
> >
> > Since we are close to the deadline, if there are still any unresolved issues or further clarifications needed, please do not hesitate to let us know. We hope that our responses and revisions meet your expectations, and if so, we would appreciate it if you could reassess your score accordingly.

---

> > > ### Comment · Reviewer_HieC · 2023-11-22
> > >
> > > Thanks for the reply. The responses addressed most of my concerns and I will raise my rating to 6.

---

> ### Author Response · Authors · 2023-11-22
> **Response to reviewer**
>
> We are grateful for your consideration of our rebuttal and thank you for your decision to raise the score. If there are any further issues or questions, we would be happy to provide additional information.

---

### Official Review · Reviewer_f7uZ · 2023-10-29

**Soundness:** 3 good
**Presentation:** 3 good
**Contribution:** 3 good
**Rating:** 8
**Confidence:** 4

**Summary:**

This paper introduces a benchmark for evaluating generalization in offline learning. Based on the benchmark, state-of-the-art offline policy learning algorithms, including BC, sequence modeling approaches, and offline RL algorithms, are tested. The results show that all the offline learning methods perform worse than online RL in both train and test environments. The results also reveal that BC is stronger to generalize to new environments than other offline learning methods.

**Strengths:**

1. This paper presents new results on the generalization of offline learning. As we know, it is important to understand the generalization ability of offline learning methods in order to apply offline methods to real-world problems. Although not very surprising, this paper first confirms that offline RL and sequence modeling approaches can be struggling to generalize to new environments.
2. The results may have a broad impact on the community. Indeed, we can no longer ignore the generalization problem of existing offline learning methods. So, more investigation is needed. The results may also have an impact on our choice of offline learning methods in application scenarios.
3. The experimental results are sufficient and convincing. In the experiments, multiple sequence modeling methods and multiple offline RL algorithms are included, two kinds of games are tested, and multiple settings are tested.
4. The new benchmark is new and an important contribution.

**Weaknesses:**

1. The paper does not discuss in depth the root causes of the generalization problem. I think it would be a great credit to the paper if the authors could share some thoughts on why.
2. There are minor problems:
- The color of the lines in Figure 2(a) is wrong.
- The results on Leaper are missing in Figure 11 and Figure 12.

**Questions:**

Can the authors share some thoughts on why offline RL does not generalize well?

---

> ### Author Response · Authors · 2023-11-17
> **Response from authors**
>
> We are grateful for your valuable feedback and hope our answers below will address your remaining concerns. We are pleased to see that you found our “results to be impactful” and “our benchmark an important contribution”.
>
> > some thoughts on why offline RL does not generalize well?
>
> We believe one reason for which offline RL methods lag behind BC is that all the offline RL approaches employ a risk-averse strategy whereby if they are encouraged to not take actions that they haven’t seen during training. However, given that we are testing generalization to new environments, **all the states the agent encounters at test time are new so offline RL methods will be averse to taking any of those actions, likely defaulting to a rather suboptimal policy.** In contrast, BC doesn’t have any such constraints so it simply uses its learned representations to decide what action is best to take by “looking at the observation closest to the test state and picking the best action”. **If BC is able to learn good enough state representations from its training data, it should be able to also learn a good enough policy that generalizes to new environments reasonably well.**
>
> On the question of **why all offline learning approaches lag behind online RL ones, we believe the reason lies in the fact that online RL collects and learns from its own data, thus seeing a much more diverse set of states than BC or offline RL approaches where the set of states is fixed by the dataset.** Given that at test time, all the states are new for the agent, having seen a wider range of states can help the agent learn better representations and have a better idea of what’s the best action in new situations. As our experiments in section 4.4 show, training on more diverse data can greatly improve the generalization of offline learning methods. Yet, as our results in section 4.5 show, simply training on data from multiple PPO checkpoints is not enough since these are sparsely sampled throughout training so they cannot span the entire space (it’s also difficult to know exactly how training dynamics play into the diversity of the data collected which could be a good direction for future work to further investigate).
>
> **We’ve updated the paper with a more in-depth discussion of this topic – see Appendix E “Discussion”**, however, we believe more work is needed to test these hypotheses in order to reach a better understanding of these results.
>
> It's important to note that our findings, which may seem surprising, underscore the significance of our benchmark. The divergence in results emphasizes the need for further research in this domain. Our results highlight the complexity of offline RL generalization and cautions against assuming that insights from Atari datasets (which is one of the most commonly used offline RL datasets) will directly transfer. Moreover, we would like to emphasize that **our results also align with other literature like [1], [2],[3] that show similar results i.e. BC > offline RL wrt generalization.**
>
> [1]: Gulcehre, C., Paine, T. L., Srinivasan, S., Konyushkova, K., Weerts, L., Sharma, A., ... & de Freitas, N. (2023). Reinforced Self-Training (ReST) for Language Modeling. arXiv preprint arXiv:2308.08998.
> [2]: Piterbarg, U., Pinto, L., & Fergus, R. (2023). NetHack is Hard to Hack. arXiv preprint arXiv:2305.19240.
> [3]: Hambro, E., Raileanu, R., Rothermel, D., Mella, V., Rocktäschel, T., Küttler, H., & Murray, N. (2022). Dungeons and Data: A Large-Scale NetHack Dataset. Advances in Neural Information Processing Systems, 35, 24864-24878.
>
>
> > There are minor problems:
> The color of the lines in Figure 2(a) is wrong.
> The results on Leaper are missing in Figure 11 and Figure 12.
>
> We thank the reviewer for pointing out these errors. We have now fixed Figure 2(a).
> **In Figures 11 and 12, our underlying dataset had 0 return, since the PPO policy itself did not perform well on that level. Hence, all the offline RL baselines too had 0 return which is the reason why the plots look empty as if the results are missing.** To clarify this, we have added a note in the captions of Figures 11 and 12
>
> We hope our responses above have successfully addressed your concerns. Please let us know if you have any outstanding questions.

---

> > ### Author Response · Authors · 2023-11-20
> > **Possible reminder**
> >
> > Dear Reviewer,
> >
> > We greatly appreciate the time and effort you've dedicated to reviewing our paper. Your insightful feedback has been invaluable in refining our work. We hope you find our responses and revisions satisfactory.
> >
> > Since we are close to the deadline, should any outstanding concerns or questions remain, we are more than willing to address them.

---

### Author Response · Authors · 2023-11-17
**Global response by the authors**

Dear Reviewers,

We hope we have addressed most of your concerns in our initial response to each reviewer.
1. We have uploaded our updated submission, with new modifications highlighted in blue font.
2. Moreover, we have updated Figure 5 to include scaling dataset size results on DT and BCT baselines as well and similar conclusions hold for them.
3. We have also updated Table 2 to highlight HPs for all the experiments that were conducted using the 10M Expert Dataset in Section 4.5. This way we make sure that we draw our conclusions only when all baselines are extensively tuned to the underlying training data distribution.
4. We also added a new section Appendix E "Discussion" where we talk about the possible reasons we hypothesize for this generalization trend: online RL > BC > offline learning.

We are grateful for the constructive feedback provided by all reviewers. Your insights have been invaluable in refining and enriching our work. We kindly request a reconsideration of the scores assigned, in light of the enhancements and clarifications we have made to the manuscript.

---

### Meta-Review · Area_Chair_Fao5 · 2023-12-03

**Metareview:**

The paper claims that current offline-RL algorithms do not perform well in RL generalization situations (e.g. ProcGen, where there is a notion of unseen "test" environments). Experimentally, the authors vary the dataset across factors such as:
* Expert or suboptimal demonstrations
* Diversity + Size of offline dataset
and show that (1) the diversity, not size of the offline dataset matters very much, and (2) even with optimal demonstrations, offline RL agents struggle significantly on test environments.

The conclusion of the paper (i.e., offline-RL algorithms can perform poorly on "test environments") is quite timely, and suggests more effort should be put into offline RL algorithms which can scale to unseen environments. All reviewers agree the paper should be accepted.

**Justification For Why Not Higher Score:**

As mentioned by reviewers, the paper focuses its evaluations primarily on ProcGen (and a relatively unknown "WebShop" environment). It would strengthen the paper considerably if the same conclusions can be seen for other standard environments, such as continuous control generalization problems (e.g. varying environmental factors for Mujoco, DM-Control).

Furthermore, the paper mostly end-to-end presents empirical insights, and does not have any theoretical improvements or contributions to cover why offline RL is suboptimal for test environments, outside of hypothesis such as offline RL being too conservative.

**Justification For Why Not Lower Score:**

All of the reviewers and I believe the conclusion (offline RL does not work yet for RL generalization) is an important statement, and should at least be accepted.

---

### Decision · Program_Chairs · 2024-01-16

Accept (poster)